



# Speciation of VOC emissions related to offshore North Sea oil and gas production

Shona E. Wilde[1], Pamela A. Dominutti[1,2], Stephen J. Andrews[1], Stephane J.-B. Bauguitte[5], Ralph R. Burton[4], Ioana Colfescu[4], James France[6,7], James R. Hopkins[1,3], Anna E. Jones[6], Tom Lachlan-Cope[6], James D. Lee[1,3], Alastair C. Lewis[1,3], Stephen D. Mobbs[4], Alexandra Weiss[6], Stuart Young[1], and Ruth M. Purvis[1,3]

[1]Wolfson Atmospheric Chemistry Laboratories, University of York, York, YO10 5DD, UK
[2]Laboratoire de Météorologie Physique, University of Clermont Auvergne, 63000, Clermont-Ferrand, France
[3]National Centre for Atmospheric Science, University of York, York, YO10 5DD, UK
[4]National Centre for Atmospheric Science, School of Earth and Environment, University of Leeds, LS2 9JT, UK
[5]Facility for Airborne Atmospheric Measurements, Cranfield University, Bedford, UK
[6]British Antarctic Survey, Natural Environment Research Council, Cambridge CB3 0ET, UK
[7]Department of Earth Sciences, Royal Holloway, University of London, Egham TW20 0EX, UK

*Correspondence to:* Ruth M. Purvis (ruth.purvis@ncas.ac.uk )

**Abstract.**

The North Sea is Europe's key oil and gas (O&G) basin with the output currently meeting 3–4 % of global oil supply. Despite this, there are few observational constraints on the nature of atmospheric emissions from this region, with most information derived from bottom-up inventory estimates. This study reports on airborne measurements of volatile organic compounds

(VOCs) emitted from O&G producing regions in the North Sea. VOC source emission signatures for the primary extraction products from offshore fields (oil, gas, condensate, mixed) were determined in four geographic regions. Measured iso-pentane to n-pentane ($iC_5/nC_5$) ratios were 0.89–1.24 for all regions, used as a confirmatory indicator of O&G activities. Light alkanes (ethane, propane, butane, pentane) were the dominant species emitted in all four regions, however total OH reactivity was dominated by unsaturated species, such as 1,3-butadiene, despite their relatively low abundance. Benzene to toluene ratios

indicated the influence of possible terrestrial combustion sources of emissions in the Southern, gas-producing region of the North Sea, seen only during south or south-westerly wind episodes. However, all other regions showed a characteristic signature of O&G operations. Correlations between ethane ($C_2H_6$) and methane ($CH_4$), confirmed O&G production to be the primary $CH_4$ source. The enhancement ratio ($\Delta C_2H_6/\Delta CH_4$) ranged between 0.03–0.18, indicating a spatial dependence on emissions with both wet and dry $CH_4$ emission sources. The excess mole fraction demonstrated that deepwater oil extraction resulted

in a greater proportion of emissions of higher carbon number alkanes relative to $CH_4$, whereas gas extraction, typically from shallow waters, resulted in a less complex mix of emissions dominated by $CH_4$. The VOC source profiles measured were similar to those in the UK National Atmospheric Emissions Inventory (NAEI) for oil production, with consistency between the molar ratios of light alkanes to propane. The largest discrepancies between observations and the inventory were for mono-aromatic compounds, highlighting that these species are not currently fully captured in the inventory. These results demonstrate





the applicability of VOC measurements to distinguish unique sources within the O&G sector and give an overview of VOC speciation over the North Sea.

# 1 Introduction

Emissions from offshore oil and gas (O&G) production have been little studied in comparison to those from onshore production.

Globally, offshore oil production accounted for around 30 % of the overall production in 2015 (EIA, 2016a). The North Sea is home to the largest number of offshore rigs worldwide with 184 operational installations as of January 2018 (Statistica, 2018). These are located across a number of different regions in the territorial waters of the United Kingdom (UK), Norway and the Netherlands. UK O&G production reached a seven year high in 2018 with an increase of more than 4 % from 2017 (Oil and Gas Authority, 2019), meaning production from the UK Continental Shelf (UKCS) met 59 % of the O&G demand of the UK

as of 2018 (Oil and Gas UK, 2019). The Norwegian sector is an evolving region of the North Sea with around 20 projects in various stages of development on the Norwegian Continental Shelf. Oil output is expected to grow by 43 % from 2019 to 2024 as production from new fields begins and older facilities are upgraded (Norwegian Petroleum Directorate). The Netherlands was the second largest producer of natural gas in the EU in 2018 (Eurostat, 2019). The onshore Groningen field is by far the largest, however production is set to cease by 2022 due to induced seismicity above the field, meaning offshore, small field

production may become increasingly important for the Dutch sector.

The release of air pollutants from O&G production has led to growing environmental and public scrutiny. Emissions of greenhouse gases such as methane ($CH_4$) are often the focus due to it's high global warming potenital (Miller et al., 2013). Interest in emissions of volatile organic compounds (VOCs) from regions of O&G production arises because of their role as precursors to tropospheric ozone (Kemball-Cook et al., 2010; Edwards et al., 2014) and secondary organic aerosols. Both of

these secondary pollutants have associated health and ecosystem effects (McKenzie et al., 2012). Detailed measurements and quantification of VOCs and their subsequent reaction products is therefore needed in order to mitigate these air quality concerns (Koss et al., 2017). Emissions of VOCs can arise at all stages of the production process, such that crude oil processing was considered to be capable of emitting around 16 % of the total global VOC emissions into the atmosphere in the late 20th century (Masnadi et al., 2018). Sources include power generation sets, compressors, pumps, flaring off excess gas, cold venting during

tank loading for transport on shuttle vessels and general fugitive emissions. Previous studies conducted in the United States have reported high VOC emissions for onshore wells and pads (Ghosh, 2018; Gilman et al., 2013; Koss et al., 2017; Simpson et al., 2010; Warneke et al., 2014; Pétron et al., 2012). These studies have shown that comprehensive VOC characterization is crucial for source identification and to aid the estimation of effects of those emissions on air quality in the surrounding regions.

The VOC composition of O&G emissions can be influenced by several variables, including the product being extracted (e.g.

oil, gas or condensate), the geological composition of the reservoir, extraction techniques, age of the rig and infrastructure components. Offshore O&G emissions are naturally more difficult to measure due to lack of access to the installations and the large number of potential sources due to the complexity of offshore extraction. Offshore platforms are different to onshore well pads and their purpose and function are extremely varied, therefore it should not be assumed that studies quantifying





onshore emissions will adequately represent emissions from the offshore sector. Few studies have examined VOC emissions from offshore facilities, whilst many of those that have follow "disaster" events, such as the Deepwater Horizon oil spill in the Gulf of Mexico in 2010 (Camilli et al., 2010; Ryerson et al., 2011) and the Elgin platform gas uncontrolled release in the North Sea in 2012 (Lee et al., 2018). Events such as these are uncommon and are therefore unlikely to be representative of
VOC emissions from the offshore sector as a whole.

Offshore emissions are explicitly identified and reported in the UK National Atmospheric Emissions Inventory (NAEI). O&G operators are responsible for the self-reporting of these emissions for each offshore production platform. Whilst emissions of $CH_4$ are allocated to individual sources, only a total mass of VOC emission is reported for each point source platform, with no information relating to emissions of individual compounds. Some estimation of the relative speciation of VOCs is made in the
inventory using historical speciation profiles, however these are generalised across source sectors and the overall uncertainties in these source profiles are not quantified. To date, VOCs in the North Sea have not been extensively studied. In this context, an improvement in the knowledge of VOC emissions from offshore O&G facilities is needed, in order to assess the potential impacts that these emissions can undergo in the atmosphere as well as improve the accuracy of emissions inventories.

Here we report measurements of VOCs made from research aircraft downwind of O&G installations over the course of
multiple research flights across all regions of the North Sea. To the best of our knowledge, this is the first time that intensive airborne VOC measurements have been made regarding O&G installations in the North Sea. Our study evaluates the general emission sources as well as a more detailed evaluation of speciation as it relates to the production of oil, gas and condensate. The source signatures of each primary extraction product were quantified using enhancement ratios to expose the spatial differences in emissions. Finally, a comparison has been made to the UK emissions inventory in order to evaluate commonalities and
discrepancies in the VOC speciation of source profiles for the O&G sector.

## 2  Methods

### 2.1  Measurement location and context

Measurements from a BAe-146 aircraft operated by the Facility for Airborne Atmospheric Measurements (FAAM) and a DHC6 Twin-Otter operated by the British Antarctic Survey (BAS) were made over a four-year period, beginning in 2015 and
ending in summer 2019. The data used here was associated with a range of different projects: Baseline, Methane Observation and Yearly Assessment (MOYA), Climate and Clean Air Coalition (CCAC) and Assessing Atmospheric Emissions from the Oil and Gas Industry (AEOG). The flights surveyed a large number of O&G installations distributed across the whole of the North Sea. Data from a total of 28 flights (approximately 128 flying hours) across multiple airborne experiments were unified to generate a single data set for this analysis and are summarised in Table A1.

The spatial distribution of emissions attributed to O&G operations was evaluated by dividing the North Sea into four distinct regions, shown by the coloured boxes in Fig. 1a. The regions were chosen for extended sampling because they contain high densities of offshore platforms and geologically distinct fossil-fuel producing reservoirs, allowing for comparison of source signatures. The North UK region comprises oil, gas and condensate producing platforms serviced from Aberdeen and Hartle-





pool. The Britannia gas field, located 130 miles north east of Aberdeen is one of the largest fields in the North Sea (Hill and Palfrey, 2003). The Norwegian Sector encompasses platforms in both the North and Norwegian Seas. Fields include Oseberg, and Asgard, which are Norway's seventh and eight biggest fields respectively. The South UK region includes a combination of fields located in the UK and Dutch sections of the North Sea. This region has the highest number of platforms. The largest field

in this region is Leman, which has a current estimated production of $5.7 \times 10^6$ m$^3$ of gas per day (Glennie, 2001). Finally, the West Shetland region is a term that incorporates platforms located in the Schiehallion, Foinaven, Clair, Alligin, Lancaster and Solan oil fields. The surveyed area lies approximately 190 km west of the Shetland Isles. Recoverable reserves are estimated to be in the region of 250–600 million barrels of oil (BP Exploration Company Ltd., 2019).

Offshore field outlines and corresponding field types were obtained from the respective O&G authorities; The Oil and Gas

Authority for UK fields (OGA), the Norwegian Petroleum Directorate for Norwegian fields (Nowegian Petroleum Directorate) and from the Geological Survey of The Netherlands for fields in the Dutch sector (NLOG). Each listed offshore field is assigned a field type of either oil, gas, condensate or a mixture. Often the dominant product of a field can change over time as reservoirs are depleted, therefore it is assumed that the field type listed is that of the most recent extraction product. Each region has distinct fossil-fuel producing reservoirs, shown in Fig. 1b. West Shetland is entirely an oil producing region, whereas the South

UK is dominated by gas production. The North UK is approximately a 50:50 mix of oil and condensate fields with a minor percentage of gas fields. The Norwegian sector contains an array of all offshore field types, which are assigned as mixed, where multiple products are extracted from the same well.

## 2.2 Instrumentation

Both aircraft were equipped with a suite of chemical and meteorological instrumentation. Basic atmospheric measurements

including thermodynamic properties, wind and turbulence were included on both aircraft, from which information about the height, stability and structure of the atmosphere can be derived. 1 Hz measurements of carbon dioxide ($CO_2$), $CH_4$ and ethane ($C_2H_6$), along with a whole air sampling (WAS) system for the collection of VOCs was available on both aircraft. Additional measurements of nitrogen oxides ($NO_x$), carbon monoxide (CO) and ozone ($O_3$) were available on some but not all of the flights. Details about FAAM instruments can be found on the FAAM website (http://www.faam.ac.uk) and Twin-Otter

(https://www.bas.ac.uk). Instrumentation and set-up of the BAS Twin-Otter aircraft is also detailed elsewhere (France et al., 2020).

### 2.2.1 Aircraft whole air samples

Discrete ambient air samples were collected in evacuated canisters via an external inlet using unique sampling systems on board each aircraft. Both systems are based on the same principles and contain comparable components. In each system,

evacuated stainless steel canisters, sealed with pneumatically actuated bellows valves (Swagelok, SS-BNVS4-C) were filled and pressurised in-flight by drawing air from the main sampling manifold using an all-stainless steel assembly double headed bellows pump. Air was continuously flushed through the internal manifold of the cases prior to filling. More specifically, the WAS system used on the FAAM aircraft (Baseline, AEOG and MOYA campaigns, Table A1) consists of sixty-four silica





passivated stainless steel canisters of three litre internal volume (Thames Restek UK) fitted to the rear hold of the aircraft. The Son of Whole Air Sampler (SWAS) is a new, updated version of the parent WAS system and was used on Twin Otter flights only (CCAC campaigns, Table A1). Cases of Silonite coated (Entech) canisters are grouped together modularly within the aircraft cabin. The SWAS has additional functionality, where cases comprise either $16 \times 1.4$ L canisters or $8 \times 2$ L canisters.

The 1.4 L version has a single valve and was filled from vacuum to 3 atm gauge pressure. The 2 L version has valves at each end and could be filled either from vacuum or by flowing through at a variably set back-pressure before filling. The latter allowed greater control for the capture of rapidly changing atmospheric events such as narrow pollutant plumes. Canisters from both systems were deployed during the campaigns and took approximately 10–20 seconds to fill, corresponding to roughly 1–2 km of horizontal flight.

## 2.2.2 Laboratory analysis of VOCs

The whole air samples were analysed post-flight using a dual-channel gas chromatograph with flame ionisation detection (GC-FID) (Hopkins et al., 2003) housed at the University of York. Aliquots of air (700 mL) were withdrawn from the sample canister and dried using a condensation finger held at -30 °C. Samples were pre-concentrated onto a multi-bed carbon adsorbent ozone precursors trap provided by Markes International Ltd. before being transferred to the columns within the gas chromatograph in

a stream of helium. The trap was held at -27 °C during sample collection and then heated to 325 °C at a rate of 20 °C s$^{-1}$ during transfer to the GC. The eluent was injected onto an aluminium oxide ($Al_2O_3$, $NaSO_4$ deactivated) porous layer open tubular (PLOT) column for analysis of VOCs. Peak identification was made by reference to a calibration gas standard containing a known amount of 30 non-methane hydrocarbons (NMHCs) ranging from $C_2$-$C_9$. Peak integration, blank correction and the application of calibration data to calculate mixing ratios was conducted using GC Soft, Inc software.

## 2.2.3 Ancillary measurements

$CH_4$ and $CO_2$ mole fractions were measured by a fast 10 Hz G2311-f Picarro (France et al., 2020) on the Twin Otter (precision at 5 sec intervals of <0.5 ppb for $CH_4$) and a Los Gatos Research Fast Greenhouse Gas Analyzer (Lee et al., 2018) on the FAAM aircraft (FGGA, precision of <2 ppb for $CH_4$). Full details of the measurement principles and the implementation of the instrument on board the aircraft, including an assessment of the instrument performance are presented elsewhere (Cain

et al., 2017; O'Shea et al., 2013). Calibration took place in-flight using standards traceable to the WMO greenhouse gas scale.

Atmospheric $C_2H_6$ was monitored by a Tunable Infrared Laser Direct Absorption Spectrometer (TILDAS, Aerodyne Research, Inc) (Yacovitch et al., 2014). This instrument applies a continuous wave laser operating in the mid-infrared region (at $\lambda = 3.3$ $\mu$m). Rapid tuning sweeps of the laser frequency result in the collection of thousands of spectra per second which are co-averaged. The resulting averaged spectrum is processed at a rate of 1 Hz using a non-linear least-squares fitting algorithm

to determine mixing ratios. A path length of 76 m is achieved using a Herriott multipass cell in order to provide the sensitivity required for a trace gas measurement. The accuracy has been tested against two standards containing ethane in mixing ratios of $39.79 \pm 0.14$ ppb and $2.08 \pm 0.02$ ppb (high concentration standard and target gas, respectively). Water vapour correc-





tions were applied within the instrument software to account for changes in humidity during the flight (Pitt et al., 2016). The instrument has a quoted precision of 50 ppt for an averaging time of 10 s.

## 2.3 Data selection

All flights took place in the daytime (between 8am - 5pm). Flight data was filtered such that only observations contained within the boundaries of the defined regions (Fig. 1) and within the planetary boundary layer (PBL) were used in further analysis. Aircraft profiles at the beginning and end of each flight were used to determine the PBL depth. PBL profiles were generally conducted upwind of the area of interest and typically spanned an altitude range of 50 ft–5000 ft. Sharp decreases in the mole fractions of $H_2O$ and $CH_4$ at a given altitude were used to indicate the PBL height, as a marker of transition to the free troposphere (typically < 500 m above sea level). Background concentrations of $CH_4$, $CO_2$ and $C_2H_6$ were determined using 1 Hz data during straight and level aircraft runs upwind of emission sources. Concentrations of pollutants were averaged over the whole run to give a value for each species per flight (Fig. A1). WAS were captured solely within the PBL on each flight. A minimum of one sample was taken upwind of the area of the area of interest in order to assess background VOC concentrations. There were two types of flight modes implemented across this dataset; survey flights and plume-targeted flights. Since some data is likely to be biased high due to the oversampling of plumes, a comparison of the absolute VOC mixing ratios is not performed here, instead the analysis is focused on hydrocarbon ratios since these should be unaffected by repeated sampling of high values. The number of samples obtained in each sampling region are shown in Table A2.

## 3 Results and Discussion

### 3.1 VOC source attribution

#### 3.1.1 Isomeric Pentane ratios

Anthropogenic emission sources have distinctive VOC signatures and therefore their mixing ratios with respect to each other can be used for source identification and characterisation. Emissions from O&G operations can specifically be identified by examining the iso-pentane to n-pentane ($iC_5/nC_5$) enhancement ratio. This ratio is defined as the slope term obtained by the linear fit of their scatter plot (Gilman et al., 2013). These species are positional isomers meaning they have similar reactivity with respect to the hydroxyl radical (OH), which is the dominant sink of atmospheric alkanes (Atkinson, 1997) and consequently similar atmospheric lifetimes of around 65 hours (Hong et al., 2019). Branched alkanes are also preferentially oxidised by nitrate radicals ($NO_3$), however the nitrate is readily photolysed by sunlight meaning concentrations are heavily suppressed during the day. Since all flights in this work took place in the daytime and transport times were of the order of a few hours, oxidation due to nitrate is assumed to have a negligible affect on the observed pentane ratios. As a result the ratio is largely independent of photochemical processing. Both species are also similarly affected by air mass mixing and dilution, therefore the ratio remains relatively unchanged during atmospheric transport and can be approximated to represent the ratio at the source of emission (Gilman et al., 2013).





The iC$_5$/nC$_5$ ratios for each region were calculated from the slope of a linear fit using reduced major axis regression (Ayers, 2001). This method is applied when the distinction between the dependent and independent variables is uncertain and deviations between fitted and observed data occur in both the $x$ and $y$ directions. Figure 2 shows the observed iC$_5$/nC$_5$ ratios for each sampling region in the North Sea. Results of the Pearson's correlation indicated that there was a significant positive association

for all regions (Norwegian sector: r(38) = 0.83, p <.001, West Shetland: r(91) = 1.00, p < .001, North UK: r(147) = 0.95, p < .001, South UK: r(338) = 0.98, p < .001). The iC$_5$/nC$_5$ ratios observed were 1.12 for the Norwegian Sector, 1.08 for West Shetland, 0.89 for the North UK and 1.24 for the South UK.

Numerous studies have been conducted characterising the iC$_5$/nC$_5$ emission ratio from both urban and O&G sources. A study by the United States Geological Survey (USGS) reported enhancement ratios for 14 different wells, finding that the isomers

were present in approximately equal amounts with ratios ranging from 1–1.28 (Ruppert et al., 2014). Similarly Simpson et al. (2010) observed a ratio of 1.10 in Alberta over oil sands mining operations. Gilman et al. (2013) reported a ratio of 0.89 at the Boulder Atmospheric Observatory (BAO), concluding that measurements at BAO are strongly influenced by O&G emissions from the Denver-Julesburg Basin (Gilman et al., 2013). These studies show that the iC$_5$/nC$_5$ emission ratio is a robust indicator of O&G emissions for onshore environments. Of more relevance to this work are studies characterising emissions ratios from

basins outside the US and in particular those offshore. Ryerson et al. (2011) reported a ratio of 0.82 for the fluid released from the Deepwater Horizon explosion in the Gulf of Mexico. Few studies exist studying O&G emissions in the North Sea, however a study into the composition of natural gas condensate from a basin in the North Sea, reported an iC$_5$/nC$_5$ ratio of 0.84 (Riaz et al., 2011).

The enhancement ratios calculated here are consistent with ratios reported in the literature for O&G emissions and signif-

icantly lower than those reported for urban and vehicular emissions (Gentner et al., 2009). The iC$_5$/nC$_5$ ratio for urban and vehicular emissions is distinctly different to the ratio from O&G emissions. Amounts of branched alkanes such as iso-pentane are increased during the refining process, therefore in processed products such as gasoline, iso-pentane is enhanced relative to n-pentane, resulting in a higher iC$_5$/nC$_5$ ratio (McGaughey et al., 2004). The highest iC$_5$/nC$_5$ ratio was observed for the Southern gas-producing region of the North Sea, suggesting a larger relative contribution of urban emissions to VOC concentrations.

Anthropogenic emissions from the UK or continental Europe are likely to influence VOC concentrations here, resulting in a higher ratio. Literature values for urban centres in the UK have been reported as 2.97 by von Schneidemesser et al. (2010) at Marylebone Road, a roadside monitoring site located in the centre of London, UK. This agrees well with the vehicular emissions ratio of 2.95 reported in Dublin, Ireland (Broderick and Marnane, 2002). The lowest ratio of 0.89, observed for the North UK is in the lower range of ratios observed for O&G emissions but is similar to the value reported for North Sea condensate,

consistent with the fact that more than 50 % of reservoirs in this region produce condensate. In summary, these results strongly indicate that the primary source of VOC emissions in the North Sea is from O&G operations.

### 3.1.2 Correlations with tracer compounds

In order to identify the specific sources of VOCs in the each region of the North Sea, emission ratios of VOCs with tracer compounds of particular sources were examined. Mixing ratios of propane are often elevated in regions of O&G production.





Light alkanes are often co-emitted in such regions and since propane is a well known tracer for O&G operations, a strong correlation with propane indicates a common source (Ghosh, 2018). Acetylene is a common tracer for combustion emissions (Fortin et al., 2005) and is therefore used in this work to identify emissions from anthropogenic urban activity. Emission ratios were calculated from the slope of a linear fit using reduced major axis regression for each region (Table 1). Species that were

recorded below the limit of detection (LOD) of the GC-FID were classed as missing and hence those that were detected in less than half the samples from each region were excluded from the analysis. This only applied to samples obtained in the Norwegian sector, where in general mixing ratios of VOCs were significantly lower than other areas of the North Sea and thus similar to the LOD. Compounds affected from this sector included pent-1-ene, trans-2-pentene, 2,3-methylpentanes, isoprene, 2,2,4-TMP and octane.

Figure 3 shows the correlation of light alkanes with propane and acetylene and the emission ratios for other species are shown in Table 1. Pearson correlation coefficients (r) and corresponding p-values were also calculated for each compound. All $C_2$-$C_5$ alkanes (ethane, n-butane, iso-butane, n-pentane, iso-pentane) showed statistically significant correlations ($p<.001$) with propane across all regions of the North Sea. These species were tightly correlated with propane ($0.94 < r < 0.99$) in the North UK and West Shetland regions, suggesting they shared a common emission source related to O&G activities. Moderate

correlations of light alkanes with propane ($0.67 < r < 0.93$) were observed in the Norwegian sector, however all other species apart from acetylene displayed weaker correlations with propane, suggesting that O&G emissions are a significant source of VOCs in this region. The South UK had weaker correlations ($0.53 < r < 0.87$) of light alkanes but stronger correlations of combustion tracers such as alkenes (ethene and 1,3-butadiene), $C_{6+}$ alkanes (hexane and heptane) and aromatic species (toluene) with propane ($0.72 < r < 0.85$), suggesting a more complex mix of emission sources, including terrestrial. Marine

traffic is a likely source of some emissions in this region due to its proximity to the UK shipping lanes, in particular the Strait of Dover, one of the busiest shipping routes in the world (European Environment Agency, 2013). A study of marine shipping emissions in China showed that alkanes and aromatics dominated the total identified VOCs from ship auxiliary engine exhausts, particularly alkanes with a carbon number greater than seven (Xiao et al., 2018). Therefore, the stronger correlations of hexane, heptane and toluene with propane in the South UK compared to other areas of the North Sea likely arise due to the influence

of shipping on VOC measurements in this region.

    Ethane was the only compound emitted in greater quantities than propane and hence emission ratios >1 were observed. Emission ratios with propane ranged from 1.18 in the North UK to 3.31 in the Norwegian sector. These emissions ratios are significantly higher than those observed by Gilman et al. (2013) (1.09) and Swarthout et al. (2013) (1.00) at BAO downwind of onshore natural gas sources. Derwent et al. (2017) reported an ethane to propane ratio of 2.4 from natural gas leakage at

Marylebone Road, London, suggesting that the high ratios observed in this work could be as a result of fugitive emissions of raw natural gas.

    Higher carbon number alkanes (2,3-methyl-pentanes) were well correlated with propane in all regions apart from the Norwegian sector, where this species is not reported. The strongest correlation was observed in West Shetland ($r(92) = 0.92$, $p<.001$). Aromatic compounds benzene and toluene were also well correlated with propane in West Sheltand, possibly representative





of the fact that this region is dominated by oil production and hence emissions of higher carbon number species are expected (Warneke et al., 2014).

In general, much weaker correlations of all species with acetylene (compared to propane) were observed in all regions (Fig. 3b and Table 1). Particularly weak correlations were seen in West Shetland (-0.07 < r < 0.53), supporting the conclusion
that O&G activities were the dominant source of VOC emissions in this region with little influence from other sources.

Propane was well correlated with acetylene in the South UK (r(357) = 0.72, p<.001) and the Norwegian sector (r(38) = 0.8, p<.001) but with high emission ratios of 4.0 and 5.62 respectively. This correlation is likely as a result of the more general combustion sources related to O&G activities, since diesel generators are widely used on drilling rigs. Aromatic compounds (benzene and toluene) were generally weakly correlated with both acetylene and propane. The strongest correlations (0.58 <
r < 0.86) of these compounds with acetylene were observed in the South UK. This suggests a common anthropogenic source since benzene is a component of vehicular and urban emissions (Baker et al., 2008) as well as a minor component of natural gas (Halliday et al., 2016).

In summary, the strong correlation with propane suggests hydrocarbon concentrations in the North Sea are primarily influenced by O&G production. The weaker correlation with acetylene suggests vehicular emissions are not a major source of these
emissions in the more Northern regions, however there is some evidence to suggest an urban and marine traffic influence on VOC concentrations in the South UK region of the North Sea.

### 3.1.3   Benzene-toluene ratio

The influence of urban emissions was further studied by utilising the benzene/toluene (B/T) emission ratio. Toluene is often co-emitted with benzene and the ratio of the two compounds is dependent on the source of emissions. Both are present in
primary vehicle exhaust emissions (Jobson et al., 2005) and from O&G sources as well as solvents, industry emissions and some natural emissions (Halliday et al., 2016; Thompson et al., 2014). Toluene has a shorter atmospheric lifetime with respect to OH, therefore the B/T ratio can be used to estimate the photochemical age of an air mass (Warneke et al., 2001). The B/T ratio can also be used to evaluate the emission sources of measured VOCs, in particular to distinguish traffic emissions from O&G emissions. Ratios in the range of 0.41–0.83 indicate emissions originating from vehicles (Langford et al., 2009).

Figure 4a shows the relationship between benzene and toluene for each region. As before, emission ratios were calculated using reduced major axis regression. Significant positive correlations (p<.001) were observed for all regions. Strong correlations were observed in the North UK, Norwegian sector and West Shetland with Pearson correlation coefficients of r(163) = 0.95, r(37) = 0.94 and r(92) = 0.94 respectively, implying these compounds have a common source. The observed B/T emission ratio was 1.29 for the North UK, 1.24 for the Norwegian sector and 1.20 for West Shetland (Table A3), suggesting vehicle emissions
are not the dominant source of these compounds since these values are higher than the typical observed range (Langford et al., 2009). These values are consistent with findings from a study at BAO by Swarthout et al. (2013) who found the lowest toluene-to-benzene (highest benzene-toluene) ratios in the north-east (T/B = 0.76 ± 0.25, B/T = 1.32 ± 0.25) sector were attributable to O&G emissions.





A much weaker, albeit significant correlation between benzene and toluene was observed in the South UK (r(357) = 0.56, p<.001). It is evident from Fig. 4a that there are two distinct segments of data with unique correlations between the two species; the first with considerably lower B/T ratios than observed in other regions of the North Sea, the second being characterised by enhancements in benzene mixing ratios and very small amounts of toluene.

Figure 4b shows the regression plot of benzene versus toluene for the South UK coloured by wind direction sector. Data with a B/T ratio between 0.41–0.83 (traffic emissions) is plotted with a diamond, accounting for 3.5 % of the observations in the Southern region. A strong positive correlation was found to exist (r(14) = 0.92, p<.001) for the traffic source and the slope obtained from the linear fit was 0.60, in the centre of the range expected for vehicle emissions (Langford et al., 2009). The traffic source was primarily observed when the wind direction was from the south or south-west, suggestive of air transported

from the UK or from continental Europe polluted by urban vehicular emissions. A similar traffic source is also visible in the North UK data, similarly exclusively observed under southerly wind conditions. There is a section of highly correlated data (r(214) = 0.81, p<.001) characterised by B/T ratios > 4 (squares, Fig. 4b), implying evidence of either an aged emission source due to the high proportion of benzene relative to toluene or an additional source of benzene that is not co-emitted with toluene. This source was dominated by air from the north-west, suggestive of aged air masses transported from the UK mainland. The

remaining fraction of data (33 %) was attributed to O&G emissions and is plotted with a triangle in Fig. 4b. This data was tightly correlated (r(125) = 0.86, p<.001) with an emission ratio of 1.12, in agreement with the range quoted by Swarthout et al. (2013) from a wind direction dominated by natural gas emissions.

### 3.2   Emission profiles of VOCs from offshore fields

Our results depict that O&G production is the dominant source of emissions in the North Sea. There are over 1000 individual

offshore fields beneath the North Sea, each listed as a specific field type (Fig. 5). To further investigate the spatial differences in emissions and to derive the VOC emission profiles from each classification of offshore field, each 1 Hz observation and each VOC measurement was spatially joined to a specific offshore field and hence field type. For each flight, the regional background of $CH_4$, $CO_2$ and $C_2H_6$ was calculated as an average of concentrations on straight and level aircraft runs upwind of any emission sources. Additionally, background VOC concentrations were calculated as the average of the lowest 1st percentile

of measurement data for each flight. Whole air samples identified as being dominated by traffic emissions using the B/T ratio (0.41 < B/T < 0.83) were removed prior to this analysis in order to more robustly compare the emissions from one field type to another.

### 3.2.1   $CH_4$ source identification

The molar enhancement ratio of $C_2H_6$ to $CH_4$ is commonly used for $CH_4$ source identification since $C_2H_6$ is emitted almost

exclusively from fossil carbon sources, whereas $CH_4$ has a mix of sources. 1 Hz measurements of $CH_4$ and $C_2H_6$ were used to characterize the $CH_4$ sources in the North Sea. In this environment, a positive correlation implies that the $CH_4$ originates primarily from O&G sources, whereas a weak to no correlation suggests biogenic or industrial sources of $CH_4$ (Ghosh, 2018). These other sources include landfills, water treatment and cattle and are only associated with very low levels of $C_2H_6$, typ-





ically < 0.2 % (Yacovitch et al., 2014). $CH_4$ from O&G sources is often co-emitted with tracers such as $C_2H_6$, resulting in considerably higher ratios ranging from 0.01–0.30 (Yacovitch et al., 2014).

The enhancement ratio ($\Delta C_2H_6$ (ppb)/$\Delta CH_4$ (ppb)) for each field type was obtained by first subtracting the average background estimates for each species during each flight from the observed enhancements over the North Sea. Any enhancement of

$CH_4$ due to an anthropogenic combustion source was removed by utilising the $\Delta CH_4/\Delta CO_2$ enhancement ratio, which is the slope of the linear regression fit of enhanced mixing ratios of $CH_4$ and $CO_2$ (Nara et al., 2014). The predominant wind direction in the UK is from the South West, meaning that our measurements likely represent emissions from both offshore platforms and onshore coastal sources. Observations at remote offshore sites show that air masses dominated by anthropogenic combustion related emissions typically have $\Delta CH_4/\Delta CO_2$ ratios of less than 20 ppb ppm$^{-1}$ (Conway and Steele, 1989). Consequently

this filter was applied to our data to remove the influence of anthropogenic emissions. Data with enhancement ratios above 20 ppb ppm$^{-1}$ was assumed to be a mix of flaring and fugitive emissions and was therefore used in further analysis to compare the signatures of individual field types (Fig. A2).

The $\Delta C_2H_6/\Delta CH_4$ enhancement ratios were calculated using reduced major axis regression and the correlations were evaluated through the calculation of the Pearson's correlation coefficient. Figure 6 shows the scatter plot of $C_2H_6$ and $CH_4$ enhance-

ments. Strong, positive correlations for field types of condensate, gas and oil were observed (r(878) = 0.98, r(1433) = 0.93 and r(3385) = 0.81) respectively with p<.001 in all cases), suggesting O&G emissions were the dominant source of $CH_4$. Mixed fields showed a weak but statistically significant correlation (r(1256) = 0.28, p<.001) along with notably smaller $CH_4$ enhancements compared to the other field types. $CH_4$ enhancements were observed with a wide range of $C_2H_6$ enhancements, with ratios ranging from 0.03–0.18. Gas fields had the lowest enhancement ratio of 0.03, indicating dry-gas emissions dominated

by $CH_4$ (Yacovitch et al., 2014). The highest emission ratio of 0.18 was observed for oil fields. Emission ratios > 0.06 have previously been observed from wet gas wells and are associated with gas containing a larger fraction of NMHCs (Yacovitch et al., 2014). Significant variability (smallest r) in the scatter of the data from oil fields was noted compared to other field types. Noticeably oil fields in the West Shetland region had a higher $C_2H_6$ content (31 %) than those in the North UK region (13 %). This could be related to the difference in the transportation methods of the extracted oil in these two regions. In the North UK,

pipelines are typically used to carry oil to shore, whereas West Shetland largely relies on shuttle tankers for oil export. VOC emissions associated with tanker loading can occur when oil is transferred from floating production storage and offloading (FPSO) vessels into shuttle tankers. During loading, light hydrocarbons dissolved in the crude oil vaporise from the liquid and accumulate in the vapour space of the tank. This increases the pressure inside the tank and therefore vapours are vented to the atmosphere (Climate and Clean Air Coalition, 2017). Additionally, high enhancement ratios > 0.85 have previously been

attributed to condensate tank flash emissions (Goetz et al., 2015), therefore the increased enhancements of $C_2H_6$ observed here possibly arise due to venting of light hydrocarbons during loading. The range of emission ratios observed across the North Sea suggests that there is a significant spatial variability in the composition of natural gas and highlights the heterogeneity of emissions across the O&G sector.





### 3.2.2 VOC composition

The whole air samples obtained during our measurements showed significant variations in VOC concentrations. Total hydrocarbon concentrations in a single sample ranged from 1.73 $\mu$g m$^{-3}$ to 155 $\mu$g m$^{-3}$, reflecting the difference between a sample in clean background air, compared to a sample captured within a plume. To broadly compare the different offshore field types,

VOCs were grouped into categories: light alkanes (C$_2$–C$_2$), heavy alkanes (C$_6$+), unsaturated and aromatic. Mixing ratios were first converted to concentrations ($\mu$g m$^{-3}$) in order to investigate which species contributed most to the total mass emitted. Figure 7a shows the contribution of the VOC categories to total VOC concentrations for each field type. Among all the species quantified, C$_2$-C$_5$ alkanes (ethane, propane, n-butane, iso-butane, n-pentane and iso-pentane) were the most abundant group for all field types, accounting for, on average, 70.2 % of the total mass emitted. The largest contribution was for oil fields

and the contributions of light alkanes in individual samples ranged from 14.0 % to 98.4 %, where ethane and propane were the dominant species. This is somewhat expected since elevated concentrations of short chained alkanes are characteristic of emissions from the O&G sector (Gilman et al., 2013). The contributions of heavy alkanes (hexane, heptane, octane, 2,3-methylpentanes, 224-trimethylpentane) was small for all field types ($<$ 8 %), suggesting O&G production was not a key source of these compounds. Aromatic and unsaturated species made similar average contributions to total VOC concentrations of 12.4 %

and 12.2 % respectively. The aromatic contribution for condensate fields was twice as high as any other field type (20.7 %). Closer inspection revealed that this was driven by particularly high aromatic contributions observed from the Erskine field, located in the North UK region of the North Sea. The variation in individual samples from this field were small compared to the field-to-field variability, suggesting the difference in emissions is largely due to different practices and equipment on individual platforms, a point also raised by Warneke et al. (2014) for onshore well pads. Contributions from unsaturated species ranged

from 7.3 % to 15.1 %. Ethene and acetylene (ethyne) were the dominant species within this group, which is suggestive of more general industrial point sources, possibly combustion generators (Washenfelder et al., 2020).

### 3.2.3 OH reactivity

Oxidation of VOCs by the hydroxyl radical (OH) to form peroxy radicals is the rate determining step in the formation of tropospheric ozone. The potential of a VOC to form ozone can be estimated by using the OH reactivity as a simple metric to

identify the key species that most readily form peroxy radicals (Gilman et al., 2013). The OH reactivity ($R_{OH-X}$) for VOCs measured in the North Sea was calculated as the product of the rate constant for the reaction of VOC "X" with OH ($k_{OH+X}$) and the VOC mixing ratio (X, molecule cm$^{-3}$) using Eq. (1) and a method outlined in Abeleira et al. (2017). Rate constants were obtained from Atkinson and Aschmann (1984) and Atkinson and Arey (2003).

$$R_{OH-X} = k_{OH+X} \left[ X \right] \tag{1}$$

Figure 7b shows the contribution of each VOC class, along with CH$_4$, to total OH reactivity for each offshore field type. Unsaturated compounds made the highest contribution to OH reactivity for all field types, contributing on average 55.5 %,





despite a low contribution of 12.2 % to total VOC concentrations. Of all the compounds in this group, 1,3-butadiene made the largest contribution to OH reactivity since it is highly reactive towards OH, making it important for ozone production despite being present in low concentrations. Light alkanes were the second most important contributor to OH reactivity for all field types. Despite being the most abundant group of compounds, contributions to OH reactivity were approximately

half that of unsaturated species, with an average contribution of 31.8 %. Contributions of these species became increasingly important in the order: gas < oil < condensate, indicating the prevalence of these compounds in emissions from oil extraction. Previous studies conducted onshore have identified alkanes to be the largest contributors to OH reactivity in regions of O&G production, with alkenes and biogenics as minor contributors due to their relatively low abundance (Gilman et al., 2013). However, Fig. 7b indicates that in the more remote offshore environment, where there are significantly less emission sources

and VOC concentrations are generally lower, OH reactivity is dominated by fast reacting species with OH. The contribution of $CH_4$ was minor for all field types, despite being the primary component of natural gas, exposing the importance of non-methane VOCs (NMVOCs) in regard to $O_3$ formation. The greatest $CH_4$ contribution was observed for gas fields (2.87 %), owing to the characteristics of dry-gas emissions, which are predominantly $CH_4$. Due to their slower reaction rates, alkanes such as $CH_4$ and ethane are likely to contribute to $O_3$ formation on larger spatial scales, rather than in the local environment

and this is potentially more significant in regions of oil or condensate production where alkanes make a heavier contribution to total OH reactivity.

### 3.2.4   Excess mole fraction

In order to compare the emission profile of VOCs measured for each field type and to provide some general quantification of emissions, the relative abundance of VOCs to $CH_4$ was calculated. The "excess mole fraction" (EMF) is defined as the relative

abundance of NMHCs to the sum of $CH_4$ and non-$CH_4$ mixing ratios (Bourtsoukidis et al., 2019). The background NMHC mixing ratios ($NMHC_{BG}$) were defined as the lowest 1 % of samples during each flight within each sampling region in the PBL and were subtracted from the total mixing ratios. Only compounds that were quantified on all flights were included in this analysis in order to keep the number of compounds making up the total NMHC concentration consistent. This resulted in the exclusion of isoprene, pent-1-ene, trans-2-pentene and 2,2,4-TMP. The excess mole fraction was calculated for each field type

using the Eq. (2):

$$EMF = \frac{\sum(NMHC - NMHC_{BG})}{([CH_4] - [CH_4]_{BG}) + \sum(NMHC - NMHC_{BG})} \times 100 \qquad (2)$$

Figure 8a shows the mean EMF calculated for each field type. Oil fields had the highest EMF of 28.3 %. This indicates that the VOC mixture emitted by oil fields consists of heavier compounds compared to gas or condensate fields. Gas fields had a much smaller amount of VOCs relative to $CH_4$ compared to oil or condensate fields, with an average EMF of 12.6 %,

representing the fact that natural gas is 70–90 % $CH_4$. This is consistent with a study in the Uintah Basin by Warneke et al. (2014) who found that oil wells had a higher VOC to $CH_4$ ratio to gas wells due to the heavier composition of hydrocarbons extracted by oil pads compared to gas. This is also consistent with Fig. 7, where oil fields had the highest percentage contri-





bution (82 %) of $C_2$–$C_5$ alkanes to total VOC concentrations. Similarly Bourtsoukidis et al. (2019) assigned high EMFs in the Arabian Peninsula to air originating from the oil fields and refineries of Iran and low EMFs to the gas fields of Turkmenistan. The EMF for condensate fields sits in the middle of those for O&G fields, with a mean value of 16.4 %. This is somewhat expected since gas condensate is a mixture of low-boiling hydrocarbon liquids and is predominantly made up of pentane isomers

with relatively small amounts of $CH_4$ or ethane (Speight, 2019). Mixed fields had the lowest mean EMF of 12.3 % but with the highest standard error of 2.55 %. Fields listed as mixed are solely located in the Norwegian sector, where the term defines reservoirs where more than one fossil fuel product is extracted over the fields lifetime. The EMFs for individual fields in this region ranged from 2.5 % to 33 % (Fig. A3) reflecting the individual characteristics of each reservoir and the differences in the types of production across this region.

The EMF can be related to water depth as displayed by Fig. 8b, which shows the relative density distributions (smoothed histogram) of water depths for platforms in the North Sea. Generally the EMF increases with increasing water depth. Gas extraction principally occurs in water depths less than 100 m, which results in low EMFs and emissions dominated by $CH_4$. North Sea condensate is typically extracted in water depths between 50 m and 200 m, resulting in an increase in emissions of light alkanes relative to $CH_4$. Oil production is more greatly associated with deep water extraction. The greatest water depths

were in the West Shetland region, with extraction taking place in water deeper than 400 m. Subsequently, when broken down by study region as well as field type, the highest EMF of 38.1 % was also observed for West Shetland, showing that deep water extraction results in emissions richer in higher molecular weight hydrocarbons relative to $CH_4$. Deepwater extraction is increasing worldwide (EIA, 2016b) and whilst one study found that deepwater facilities had moderate emission rates compared to shallow water sites (Yacovitch et al., 2020), the difference in the composition of emissions could be significant in terms of

$O_3$ production, since longer chained alkanes have shorter lifetimes with respect to OH than $CH_4$. This work shows that the EMF can be a useful tool in separating emissions from oil, gas or condensate exploitation and supports the conclusion that wet natural gas contains a more complex mix of NMHCs than dry natural gas.

## 3.3 Comparison of VOC speciation to emission inventories

The UK NAEI is the primary source of inventory information for the UK. The inventory provides pollutant specific gridded

emissions at a 1 km x 1 km resolution. Emissions are split into source sectors such as road transport, waste, agriculture and offshore. Point source emissions are also included, such as individual offshore platforms or power plants. VOCs are generally represented in the form of total NMVOC, with the exception of carcinogenic compounds benzene and 1,3-butadiene, which are quantified in the point source inventory. In order to extract specific VOC emissions estimates, the total NMVOC estimate can be combined with source profiles from Passant (2002). This includes a series of NMVOC species profiles which describe the VOC

composition from each source, given as a percentage contribution to total NMVOC emissions. The speciation profiles include over 600 different compounds from around 250 different sources. For each year in the NAEI the profiles are held constant and are applied to a new total VOC inventory. For well known sectors, constant profiles are assumed to be a reasonable approximation. However, for less well known sources, this could introduce uncertainty into emissions estimations, specifically





in the context of modelling tropospheric ozone as the results are sensitive to the VOC speciation, which is used as model inputs. Incorrect speciation means it is difficult to accurately assess the impact of emissions.

In terms of offshore VOC speciation, four relevant profiles from Passant (2002) were identified. These are listed as crude oil production, crude oil distribution, industrial combustion of gas and natural gas flares. Multiple sources are represented by a

single profile meaning these profiles are used to represent the entire offshore sector, including emissions from flaring, venting, loading and storage. Figure 9 compares the emission ratios of VOCs to propane for these profiles to the North Sea measurements from this work. (Note the flaring profile was not included here due to the lack of common species between measurements and inventory.) All offshore field types show a consistent trend in Fig. 9 with the VOC to propane ratio generally decreasing with increasing molecular weight, highlighting how O&G emissions are dominated by light alkanes. However the absolute values

vary significantly, again exposing the non-uniform nature of emissions from the O&G sector.

Our measurements closest resemble the oil production VOC profile from the NAEI, shown in Fig. 9, particularly for emission ratios of some light alkanes (n-butane, n-pentane, iso-butane), which are both qualitatively and quantitatively consistent. Obvious differences are observed for species with a carbon number greater than 5. The enhancement ratio for hexane is enhanced in the oil production profile compared to our measurements, with the largest discrepancy seen for condensate fields. The opposite

is true for the mono-aromatic species (benzene and toluene), where the enhancement ratio in the inventory profile is significantly lower than the measurement data for benzene and emissions of toluene are not included at all, suggesting the inventory may be lacking in some information. The gas combustion profile appears to capture similar trends to the measurements and also includes aromatic compounds benzene and toluene. However, this profile shows a higher contribution of VOCs relative to propane compared to observations for all species, potentially leading to the overestimation of some species should these

profiles be used in the estimation of individual VOC emissions. The oil distribution profile is sparse in terms of the number of species reported, only including alkanes up to $C_5$ and no mono-aromatics. In addition, the quantitative values of emission ratios are dissimilar to the both the observations and the other inventory profiles, suggesting this profile does not well represent offshore O&G emissions.

This work shows that VOC emissions are unique to their source and therefore using a single profile to representative multiple

emission sources will likely mean emissions are poorly described in the inventory. We recognise however that the measurements made here represent an accumulation of emissions from all potential sources located on offshore platforms, therefore it is unlikely that any one source-specific profile will agree entirely. Despite this, it is clear that the profiles are not inclusive of all species and that considerable variability exists in emissions from the O&G sector, which is not currently reflected in the inventory. In order to assess the accuracy of the NAEI source profiles, more detailed study into specific sources and activities

is needed.

## 4   Conclusions

This study presents an overview of VOCs emitted from O&G operations in the North Sea, which is a rarely investigated but rapidly changing region. Data from 28 research flights were aggregated to provide a representative picture of the spatial





distribution of VOCs across all regions of the North Sea. Enhancement ratios between pentane isomers identify O&G activities to be the dominant source. Characteristic enhancements of iso-pentane over n-pentane of 0.89, 1.08, and 1.12 in the North UK, West Shetland and Norwegian sector respectively, are consistent with literature values identifying emissions from O&G activities. A ratio of 1.24 ppb observed in the South UK provides evidence of an urban influence on emissions since branched

isomers are more prevalent in refined products such as petrol. Enhancement ratios of individual VOCs with specific tracer compounds were utilised to determine the contribution from unique emission sources. Propane was used as an O&G tracer while acetylene was used to identify vehicular emissions. Strong correlations of light alkanes with propane and generally weak correlations with acetylene confirmed that hydrocarbon concentrations in the North Sea are primarily influenced by O&G production. Emissions originating from urban traffic sources were identified in the South UK through use of the benzene to

toluene enhancement ratio, where approximately 4 % of data from this region was characterised by a B/T ratio of 0.6, typical of traffic emissions (Langford et al., 2009).

    The source profiles of offshore field types were investigated in terms of the primary product; oil, gas, condensate or a mix of them. The $C_2H_6$ to $CH_4$ enhancement ratio highlighted the significant spatial variability in the composition of emissions from offshore O&G production. Ratios ranging from 0.03 to 0.18 indicated "wet" emission sources containing high ethane content

as well as "dry" emissions primarily composed of $CH_4$. The distribution of individual VOCs was similar for all field types, with $C_2$–$C_5$ alkanes being the dominant species, however the relative contribution of VOCs to $CH_4$ was unique to each extraction product. The importance of studying VOCs in addition to $CH_4$ was exposed through calculations of VOC-OH reactivity, which showed unsaturated species, followed by light alkanes were the most important in terms of local $O_3$ formation. $CH_4$ contributed less than 3 % despite it's dominance in terms of emissions from this sector. Through calculation of the excess mole

fraction, the diversity in emissions from O&G activities was emphasized. Deep water oil extraction resulted in emissions of hydrocarbon-rich, associated gas, whereas gas extraction in shallow waters yields emissions of $CH_4$-rich, non-associated gas. EMFs typically increased with water depth, suggesting that emissions of VOCs from O&G activities may become increasingly important relative to $CH_4$ in terms of their impact on air quality as current reservoirs are depleted and exploration shifts to more challenging, deeper waters.

A comparison of our results to the source profiles found in the UK NAEI revealed several discrepancies in terms of relative speciation. The VOC to propane ratio for common species was compared to profiles relating to gas combustion, oil production and oil distribution. Whilst the profile for oil production agrees fairly well with measured molar ratios of light alkanes, deviations occurred for the higher carbon number hydrocarbons, particularly hexane, which was higher relative to propane in the inventory compared to measurements. Considerable differences were also seen for benzene and the absence of other aromatic

species and alkenes in the inventory profiles suggest these sources are not completely characterised in the current inventory, although the overall mass of emissions may still be correct.

    Overall, this work uses novel and unique data to provide a better understanding of a relatively understudied source of emissions from North Sea O&G production, which has the potential to impact local and regional air quality. The VOC speciation profiles established here could be used to update the current inventory, providing a set of observational-based profiles specific

to each fossil fuel product.



*Data availability.* Data can be obtained upon request from the authors.

*Author contributions.* Manuscript drafted, data analysed and figures produced by SW with help from PD, RP, JL, JH, AL. Experimental design and flight planning by JL, RP, SW, SM, RB, IC. Aircraft set-up and in flight measurements performed by SW, PD, SA, SB, JF, AJ,

TLC, AW, JL, SY, RP. Laboratory measurements made by SW, PD, JH. Reviewing and editing by AL, JH, JL, JF, RP.

*Competing interests.* The authors declare no competing interest.

*Acknowledgements.* Airborne data was obtained using the BAe-146-301 Atmospheric Research Aircraft (ARA) flown by Airtask Ltd. and managed by FAAM Airborne Laboratory, jointly operated by UK Research and Innovation (UKRI) and the University of Leeds and a DCH6 Twin-Otter operated by the British Antarctic Survey (BAS). The FAAM aircraft data was collected as part of the Demonstration

Of A Comprehensive Approach To Monitoring Emissions From Oil and Gas Installations (AEOG) project (Reference: NE/R01454X/1) and Improved understanding of accidental releases from oil and gas industries offshore project (Reference: NE/M007146/1) funded by the Natural Environment Research Council (NERC). The Twin-Otter aircraft campaigns were funded under the Climate and Clean Air Coalition (CCAC) Oil and Gas Methane Science Studies (MSS), hosted by the United Nations Environment Programme. Funding was provided by the Environmental Defense Fund, Oil and Gas Climate Initiative, European Commission, and CCAC. We acknowledge the Offshore Petroleum

Regulator for Environment and Decommissioning (OPRED) and Ricardo Energy & Environment for their involvement as project partners on the AEOG project.





**Table 1.** Emission ratios from the slope of the linear fit using reduced major axis regression and $r^2$ values for tracer compounds with listed VOCs in each sampling region. Values are only shown for compounds which had correlations with tracer compounds with an $r^2 > 0.4$.

| Compound | Acetylene (ppb ppb$^{-1}$) $ER_{C_2H_2}$ | $r^2$ | Propane (ppb ppb$^{-1}$) $ER_{C_3H_8}$ | $r^2$ |
|---|---|---|---|---|
| **North UK** | | | | |
| Ethane | — | — | 1.181 | 0.943 |
| iso-Butane | — | — | 0.117 | 0.962 |
| n-Butane | — | — | 0.294 | 0.977 |
| iso-Pentane | — | — | 0.055 | 0.887 |
| n-Pentane | — | — | 0.061 | 0.976 |
| trans-2-Pentene | 0.008 | 0.653 | — | — |
| Pent-1-ene | — | — | 0.004 | 0.611 |
| 2,3-Methylpentanes | — | — | 0.012 | 0.510 |
| Hexane | — | — | 0.013 | 0.749 |
| **Norwegian Sector** | | | | |
| Ethane | — | — | 1.181 | 0.943 |
| iso-Butane | — | — | 0.117 | 0.962 |
| n-Butane | — | — | 0.294 | 0.977 |
| iso-Pentane | — | — | 0.055 | 0.887 |
| n-Pentane | — | — | 0.061 | 0.976 |
| trans-2-Pentene | 0.008 | 0.653 | 0.000 | 0.484 |
| Pent-1-ene | — | — | 0.004 | 0.611 |
| 2,3-Methylpentanes | — | — | 0.012 | 0.510 |
| Hexane | — | — | 0.013 | 0.749 |
| **Norwegian Sector** | | | | |
| Ethane | — | — | 3.308 | 0.705 |
| Propane | 5.624 | 0.640 | — | — |
| iso-Butane | 1.788 | 0.581 | 0.318 | 0.856 |
| n-Butane | — | — | 0.421 | 0.717 |
| Acetylene | — | — | 0.178 | 0.640 |
| But-1-ene | -0.071 | 0.662 | -0.013 | 0.462 |
| iso-Pentane | 0.770 | 0.503 | 0.139 | 0.563 |
| n-Pentane | — | — | 0.243 | 0.452 |
| Heptane | — | — | 0.072 | 0.578 |
| Benzene | — | — | 0.098 | 0.411 |
| **West Shetland** | | | | |
| Ethane | — | — | 2.481 | 0.990 |
| iso-Butane | — | — | 0.432 | 0.987 |
| n-Butane | — | — | 0.650 | 0.986 |
| iso-Pentane | — | — | 0.278 | 0.963 |
| n-Pentane | — | — | 0.258 | 0.957 |
| trans-2-Pentene | — | — | 0.006 | 0.497 |
| 2,3-Methylpentanes | — | — | 0.099 | 0.843 |
| Hexane | — | — | 0.094 | 0.479 |
| Isoprene | — | — | 0.067 | 0.758 |
| Benzene | — | — | 0.031 | 0.425 |
| Toluene | — | — | 0.021 | 0.680 |
| **South UK** | | | | |
| Ethene | 1.857 | 0.651 | 0.465 | 0.556 |
| Propane | 3.997 | 0.515 | — | — |
| iso-Butane | 1.585 | 0.455 | 0.397 | 0.632 |
| n-Butane | — | — | 0.584 | 0.758 |
| Acetylene | — | — | 0.250 | 0.515 |
| iso-Pentane | 1.112 | 0.433 | 0.279 | 0.654 |
| 1,3-Butadiene | 0.167 | 0.823 | 0.037 | 0.519 |
| 2,3-Methylpentanes | 0.235 | 0.651 | 0.059 | 0.524 |
| Hexane | 0.246 | 0.603 | 0.062 | 0.714 |
| Heptane | 0.209 | 0.603 | 0.052 | 0.529 |
| 224-TMP | 0.148 | 0.493 | — | — |
| Toluene | 0.669 | 0.731 | 0.167 | 0.602 |



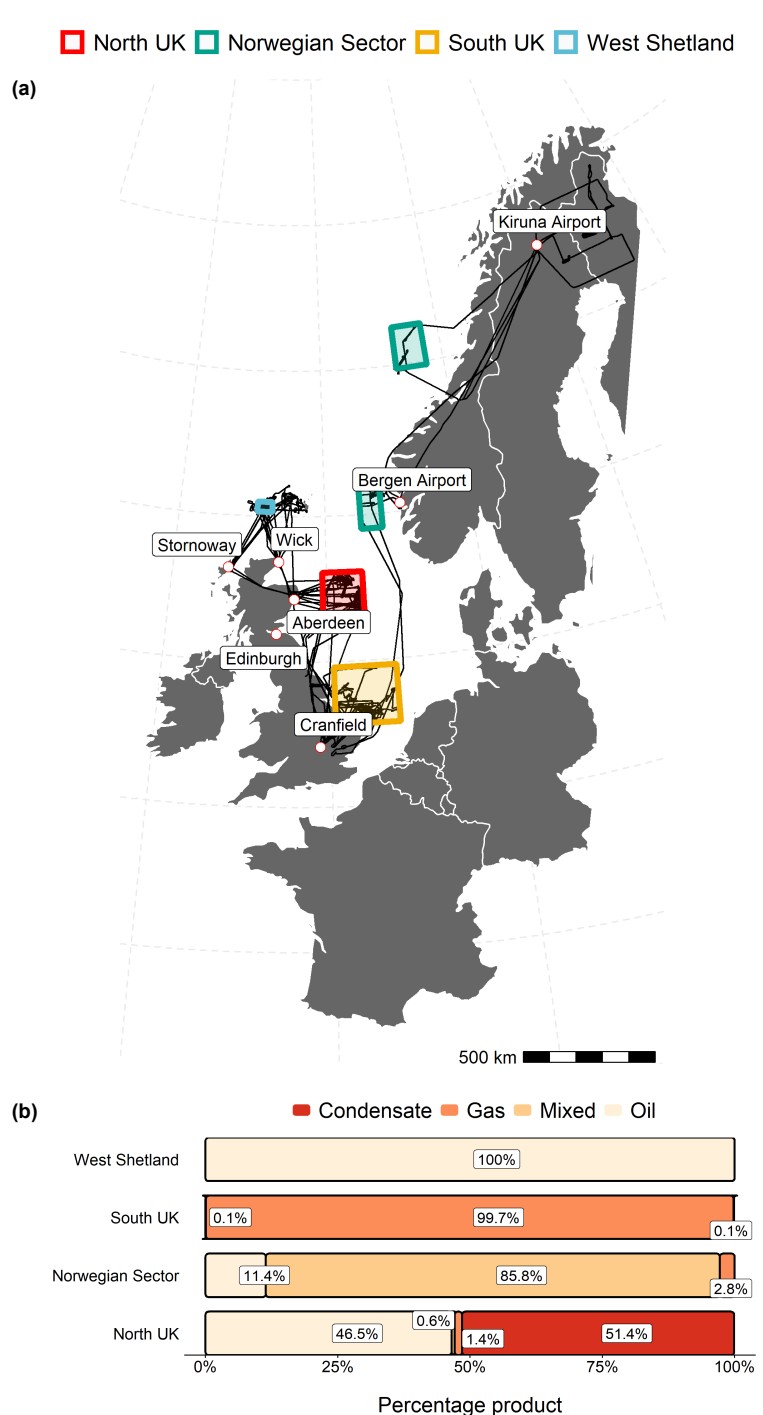

**Figure 1.** (a) Regions of the North Sea defined for analysis. The black lines represent the flight tracks of the research aircraft. (b) Percentage contribution of each offshore field type to the total area of all fields in each region. Country polygons were obtained using the **rnaturalearth** R package (South, 2017).



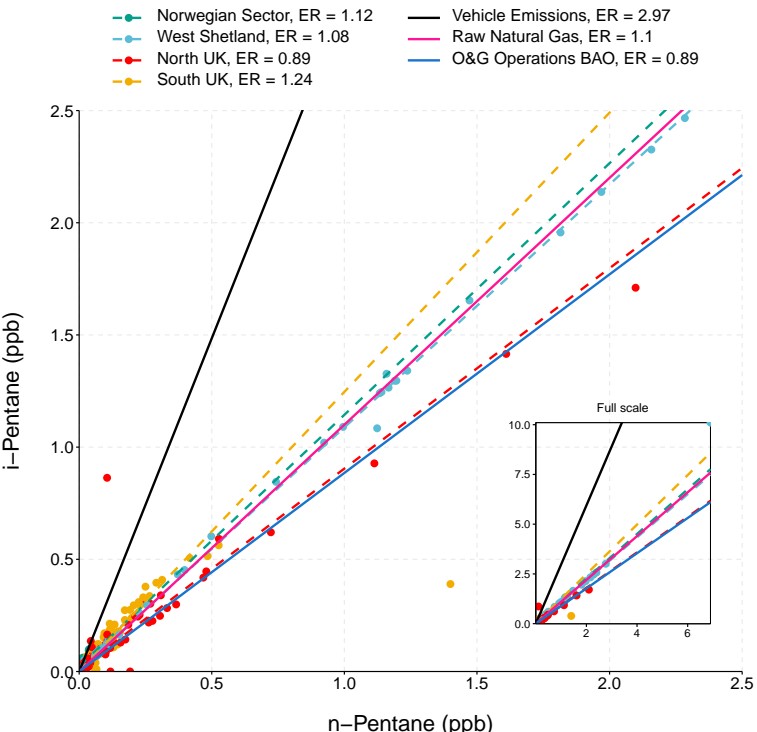

**Figure 2.** Scatter plot of iso-pentane and n-pentane observations in each sampling region. Dashed lines indicate the linear fit for each region obtained by reduced major axis regression. The solid black line indicates the ratio from vehicular emissions (von Schneidemesser et al., 2010), the solid pink line shows the ratio of raw natural gas in the North Sea obtained from the Department for Business, Energy and Industrial Strategy (BEIS) and the solid blue line indicates a typical ratio from O&G emission sources (Gilman et al., 2013). Inset shows the full range of observations.



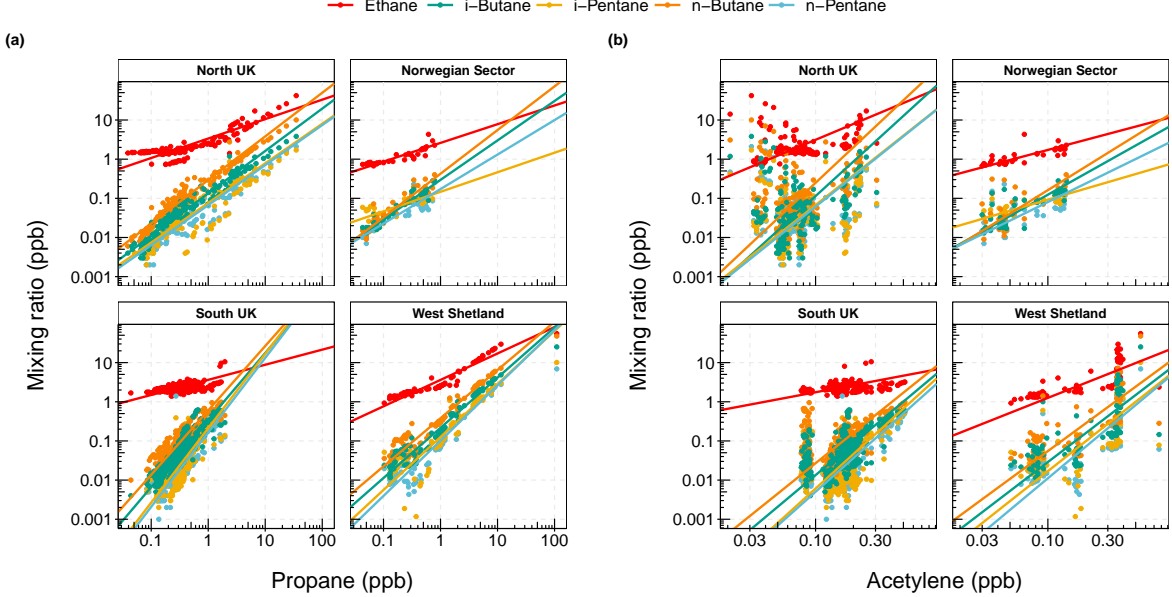

**Figure 3.** Correlation of light alkanes with (a) propane and (b) acetylene in each sampling region. The solid lines represent the linear fit using reduced major axis regression. Note the log scale used on both axis.

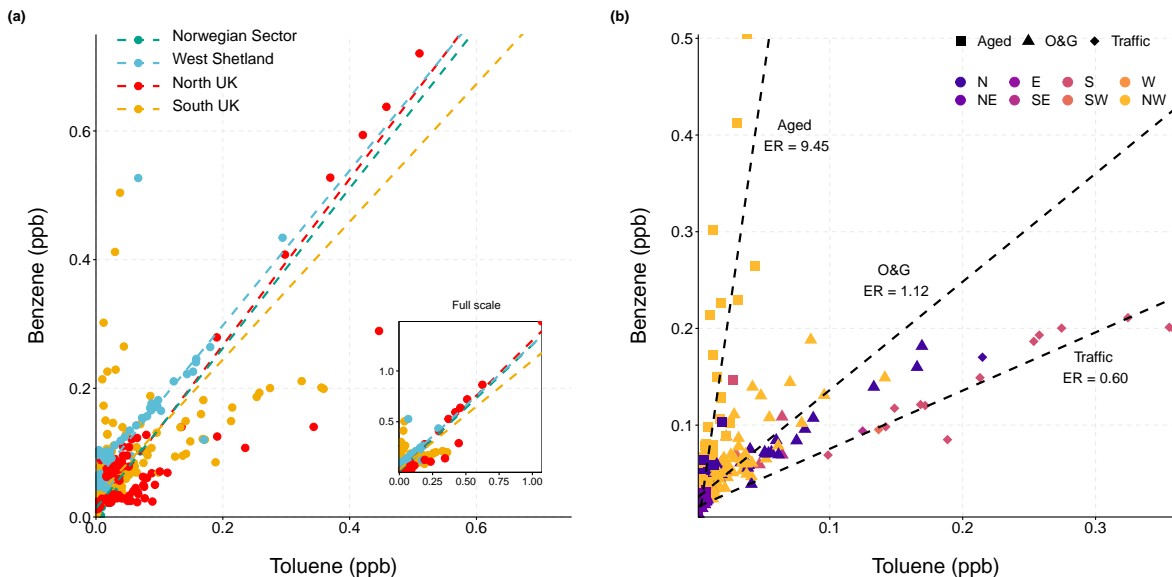

**Figure 4.** (a) Scatter plot of benzene versus toluene for all sampling regions. Inset shows the full scale of observations. (b) The South UK only, where the colour represents the average wind sector during the time the sample was taken and the shape represents identified emission sources. In both figures the dashed lines indicate the linear fit obtained by reduced major axis regression.





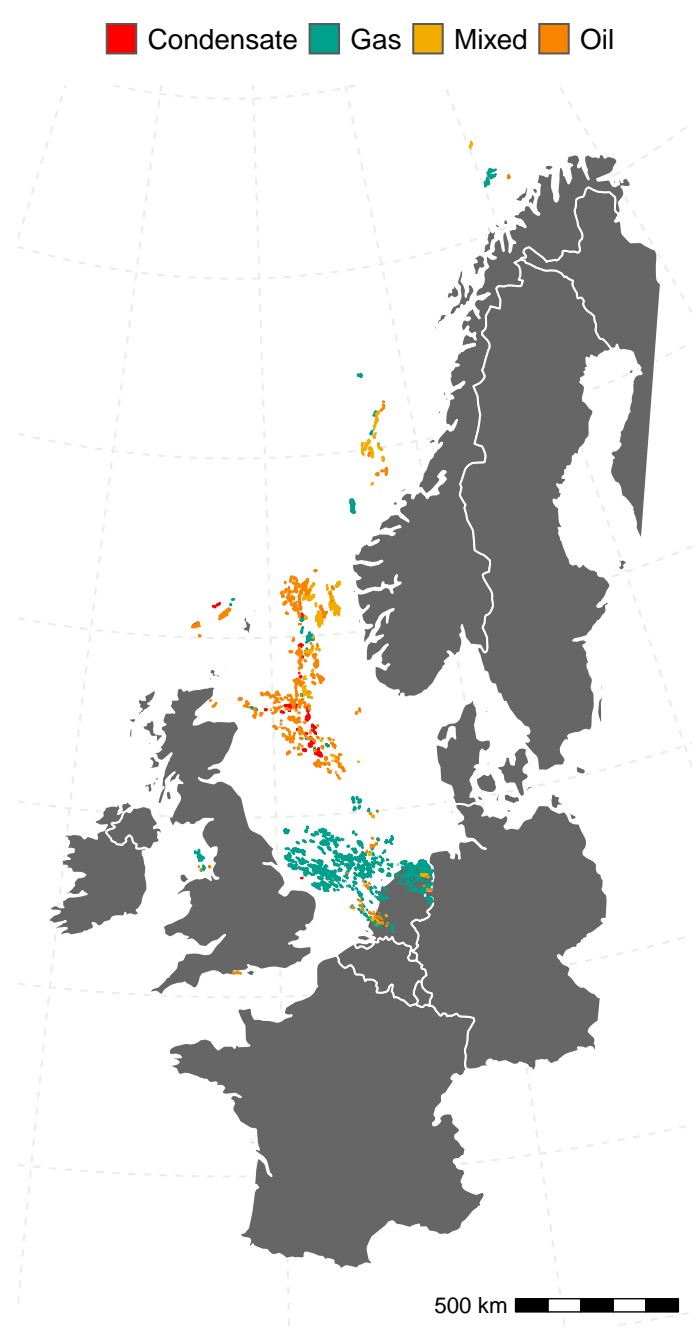

**Figure 5.** Location of all offshore fields in the North Sea. Each polygon is coloured by the extraction product from each field. Country polygons were obtained using the **rnaturalearth** R package (South, 2017).

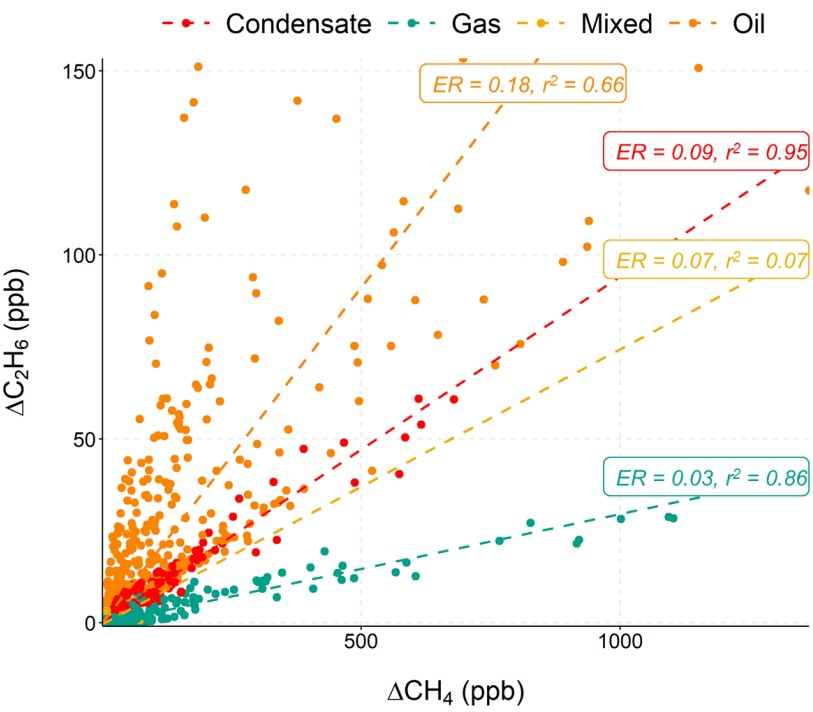

**Figure 6.** Scatter plot of $\Delta C_2H_6$ vs $\Delta CH_4$ for each offshore field type. Observations were filtered to those with a $\Delta CH_4/\Delta CO_2$ ratio greater than 20 ppb ppm$^{-1}$ as these were considered to be uninfluenced by anthropogenic urban emission sources. Dashed lines indicate the linear fit obtained from reduced major-axis regression.



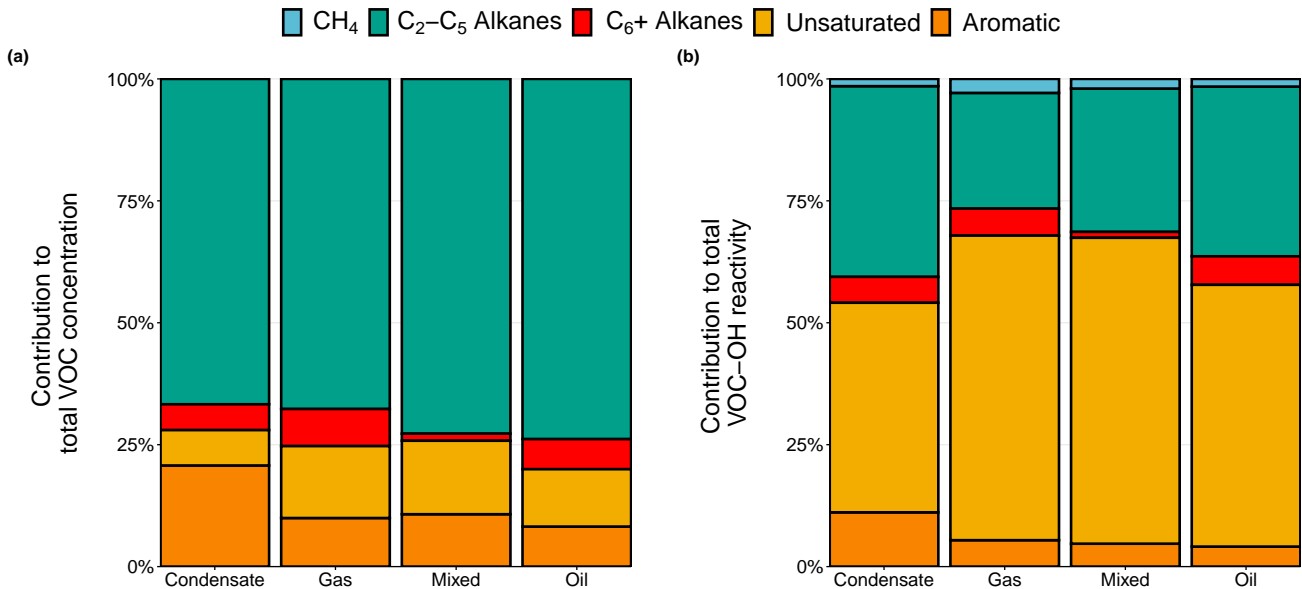

**Figure 7.** Percentage contribution of different VOC categories to (a) total VOC concentrations by mass and (b) total VOC-OH reactivity measured for each field type. The contribution of $CH_4$ is shown only for OH reactivity to demonstrate the relative importance of the other VOCs in terms of $O_3$ formation.





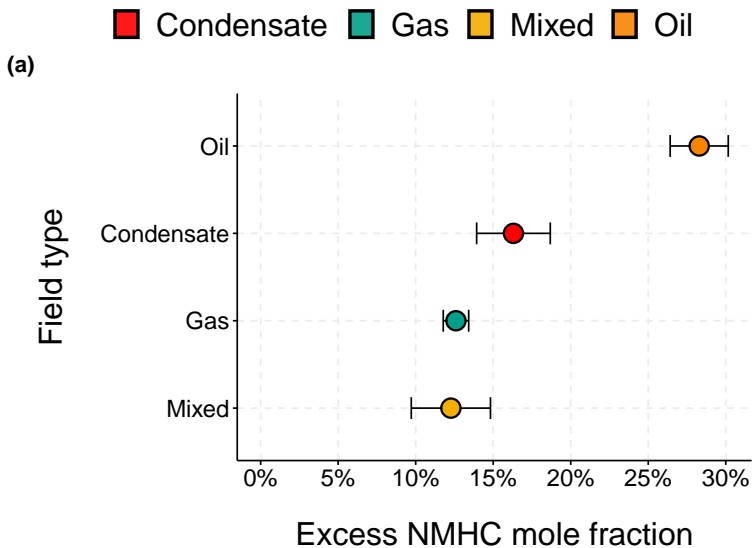

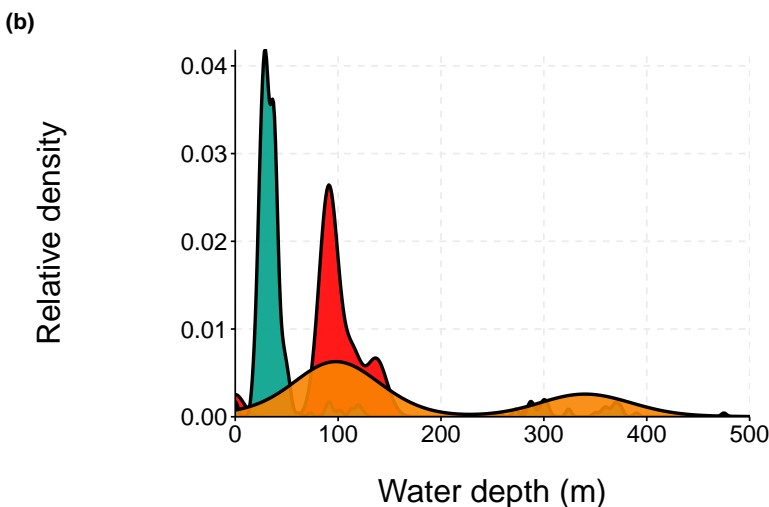

**Figure 8.** (a) Mean excess mole fraction for each offshore field type. Error bars represent one standard error. (b) Smoothed density distribution of water depth obtained for each offshore field in the North Sea, coloured by field type. A depth of 305 m (1000 ft) defines deepwater.



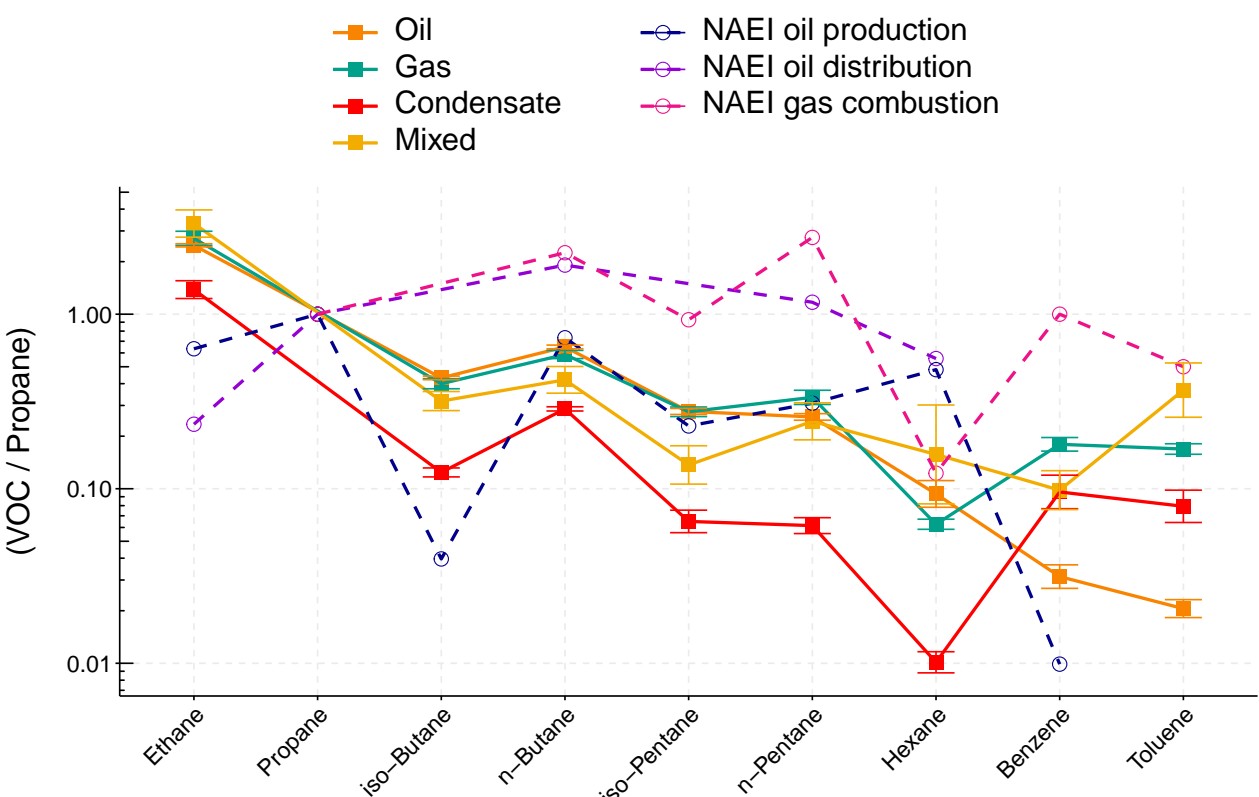

**Figure 9.** Emission ratios of VOC with propane calculated by reduced major axis regression for each offshore field type (note the log scale). Error bars represent the 95 % confidence intervals. The dashed lines show the ratio obtained from the NAEI speciation profiles in Passant (2002) for oil production, oil distribution and gas combustion.





**Table A1.** Summary of flight data used in this study. MOYA = Methane Observation and Yearly Assessment, CCAC = Climate and Clean Air Coalition, AEOG = Assessing Atmospheric Emissions from the Oil and Gas Industry. Baseline refers to a set of initial flights conducted in 2015 to serve as a baseline comparison in future work. Regions are those defined in Figure 1.

| Flight number | Date | Campaign | Region |
|---|---|---|---|
| B907 | 2015-05-13 | Baseline | South UK |
| B908 | 2015-05-20 | Baseline | South UK |
| B910 | 2015-05-26 | Baseline | South UK |
| B912 | 2015-06-09 | Baseline | South UK |
| B913 | 2015-06-23 | Baseline | North UK |
| B918 | 2015-07-23 | Baseline | North UK |
| C095 | 2018-04-19 | CCAC | South UK |
| C096 | 2018-04-20 | CCAC | South UK |
| C099 | 2018-04-25 | AEOG | North UK |
| C100 | 2018-04-26 | AEOG | West Shetland |
| C102 | 2018-04-27 | AEOG | North UK |
| C112 | 2018-07-26 | AEOG | North UK |
| C118 | 2018-09-11 | AEOG | West Shetland |
| C119 | 2018-09-11 | AEOG | North UK |
| C120 | 2018-09-12 | AEOG | West Shetland |
| C121 | 2018-09-14 | AEOG | North UK |
| C147 | 2019-03-04 | AEOG | North UK |
| C148 | 2019-03-05 | AEOG | West Shetland |
| C149 | 2019-03-06 | AEOG | West Shetland |
| C150 | 2019-03-07 | AEOG | West Shetland |
| B325 | 2019-04-30 | CCAC | South UK |
| B327 | 2019-05-02 | CCAC | South UK |
| B328 | 2019-05-03 | CCAC | South UK |
| B329 | 2019-05-06 | CCAC | South UK |
| C191 | 2019-07-29 | MOYA | Norwegian Sector |
| C193 | 2019-07-30 | MOYA | Norwegian Sector |
| C197 | 2019-08-02 | MOYA | Norwegian Sector |
| C198 | 2019-08-02 | MOYA | Norwegian Sector |





**Table A2.** Number of WAS obtained in each defined region.

| Sampling region | Number of samples |
| --- | --- |
| Norwegian Sector | 40 |
| West Shetland | 103 |
| North UK | 166 |
| South UK | 359 |

**Table A3.** Benzene to toluene emission ratios for each sampling region obtained from the slope of the linear fit by reduced major axis regression and corresponding $r^2$ values.

| | B/T (ppb ppb$^{-1}$) | |
| --- | --- | --- |
| **Area** | ER | $r^2$ |
| North UK | 1.292 | 0.907 |
| Norwegian Sector | 1.243 | 0.880 |
| South UK | 1.071 | 0.312 |
| West Shetland | 1.203 | 0.888 |

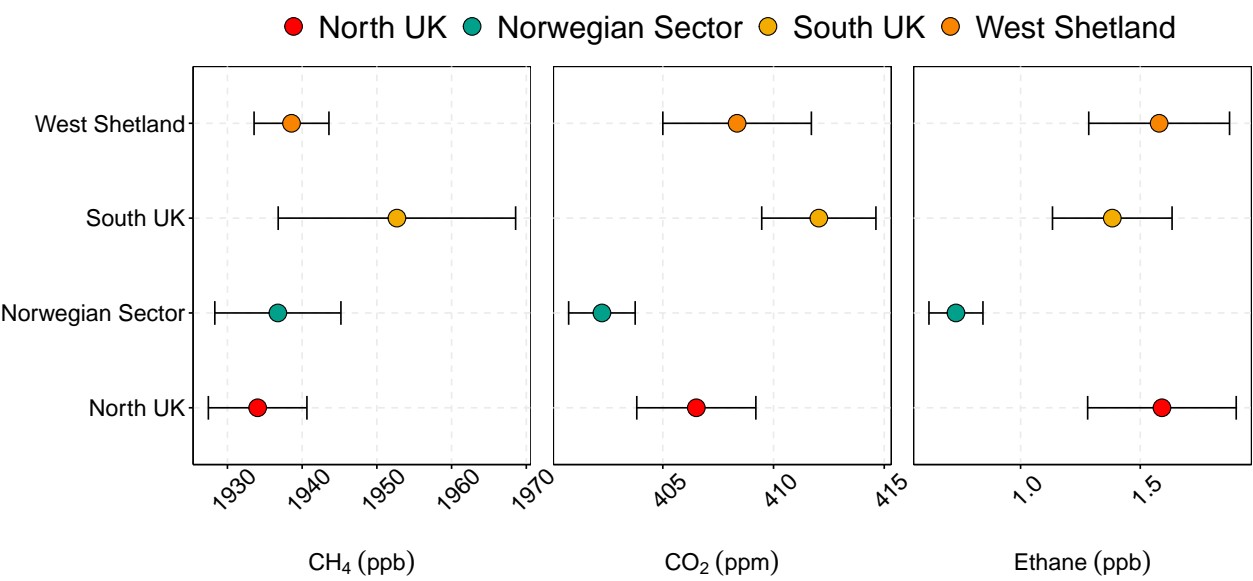

**Figure A1.** Mean background mixing ratios of $CH_4$, $CO_2$ and ethane from all flights in each sampling region. Error bars represent one standard error.

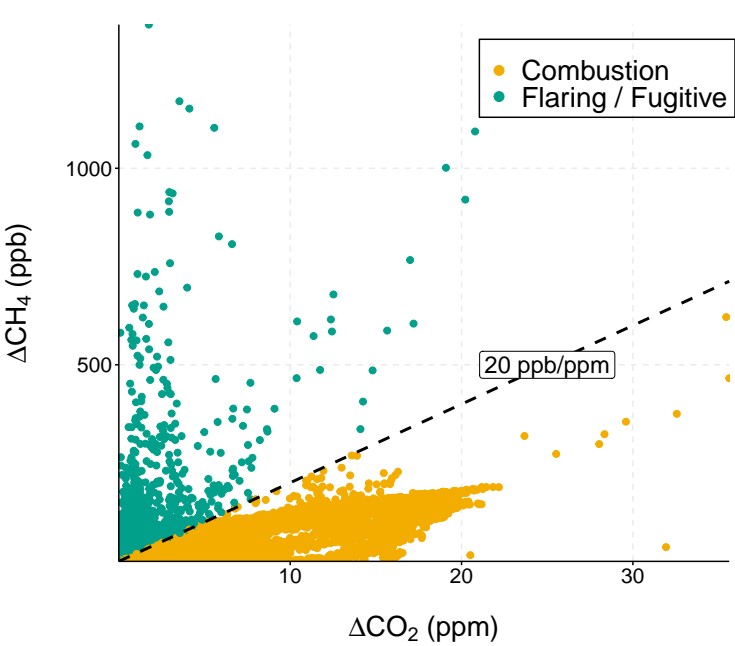

**Figure A2.** Scatter plot of $\Delta CH_4$ versus $\Delta CO_2$. The dashed line shows an emission ratio of 20 ppb ppm[-1] which was used for filtering data to remove the influence of anthropogenic emissions.



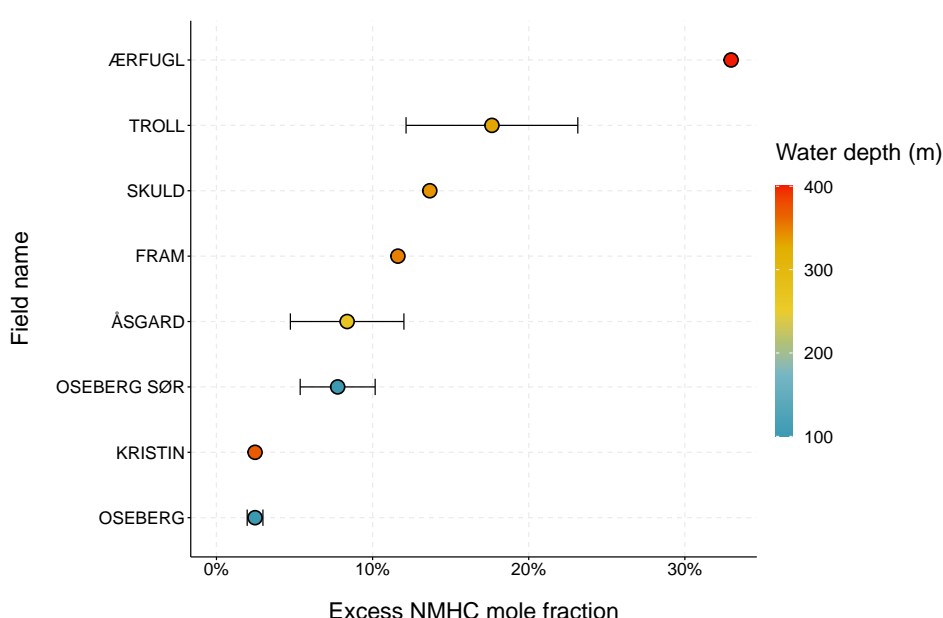

**Figure A3.** Mean excess mole fraction for individual offshore fields in the Norwegian sector of the North Sea coloured by water depth. Error bars represent one standard error.





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
