# Peer review of "Speciation of VOC emissions related to offshore North Sea oil and gas production"

_Atmospheric Chemistry and Physics, 2020_

## Referee Comment (RC1) · Jessica Gilman (Referee) · 20 Dec 2020

The authors present a thorough analysis of airborne measurements of VOCs and other trace gases over four different regions of oil and natural gas production in the North Sea. Offshore measurements of routine oil and natural gas operations are relatively few making these observations highly valuable to the scientific and regulatory communities. The authors use the iso- to n-pentane and the benzene to toluene ratios to differentiate oil and natural gas sources from urban fossil fuel sources, a minor source for the South UK sector. Tight correlations of nearly all hydrocarbons with propane (a common component of natural gas) over acetylene (a combustion tracer) further indicate the prevalence of oil and natural gas operations as the source of the light alkanes and aromatics in these regions. The authors also calculate the ethane to methane

ratios and the excess mole fraction to differentiate the VOC composition in the different source regions and to investigate the VOC composition as a function of type of product produced (oil, gas, condensate, or mix) and water depth. Additional analysis includes calculation of the OH reactivity to identify important contributors to potential ozone formation as well as comparisons to emission inventories.

The measurements are of high quality, the analysis is thorough, and the paper is very well written. I suggest publication with only a few minor technical corrections.

Corrections to be addressed:

- Impressively, I only found one typo. Page 15, Line 24. Change "representative" to "represent."

- Figure 8b: It would be best to change these to colored lines rather than the "fill to zero" as you can't see what is beneath the orange-colored oil curve.

Suggestions for the authors to consider:

- It would be helpful if Figures 1 and 5 were combined.

- It would be helpful if the colors on Figures 1b and 5 were consistent. Additionally, the orange and yellow colors used throughout were often hard to differentiate. Perhaps a darker saturation on the orange color or changing one to a different color would help the reader tell the colors apart more easily.

General comments for discussion (mainly my own curiosity):

- Did the authors measure cycloalkanes? These are often prevalent in crude oil and raw natural gas.

- Figure 4b and Line 13. I agree that either case could be true, but the slope is more suggestive of an additional benzene source as the mixing ratios for benzene are the largest for nearly all regions studied. Chemical aging would also occur along with dilution but these samples look fairly concentrated by comparison. Were any other

VOCs enhanced in the samples with enhanced benzene? It could help point to a more specific source. Benzene is used in glycol dehydrators, a common piece of equipment in US gas fields. I wonder if similar equipment is used in these fields and could be the source of benzene-only emissions other than solvent usage.

- Are the authors planning to calculate emission fluxes of methane and VOCs using these measurements in a future analysis? It would be great if so.

---

## Referee Comment (RC2) · Anonymous Referee #2 · 23 Dec 2020

**Speciation of VOC emissions related to offshore North Sea oil and gas production by Wilde et al.**

This paper will be an important contribution to the literature. It addresses the fact that there are "few observational constraints on the nature of atmospheric emissions from this region". This region has a large number of oil and gas operations within it and as such it is presumed that there are significant emissions from these processes. The paper characterizes those emissions from a large number of research flights encompassing over 120 flight hours and the paper divides the study area into four different sectors that have geologically distinct fossil fuel reservoirs. Whole air samples were collected and then subsequently analyzed for a range of NMHCs (C2-C8). Methane and ethane were also measured on-board and these are discussed in the paper as well.

The paper is well written and a large amount of high quality work went into conducting the flights, analytical work, data analysis, and synthesis and I applaud the author's efforts. In my view the paper could be improved by making it a bit more concise. At times, I found it rambling and overly speculative.

I like Figure 2 as it nicely shows that the four different sectors have i-pentane to pentane ratios that are indicative of oil and gas operations influence. Somewhere I would state that emissions from sources other urban and O & G are not expected in this region (e.g., fires). In section 3.1.2 there is a discussion about correlations with tracer compounds. Here in a lead up to discussing Figure 3 it is stated that mixing ratios of propane are often elevated in regions of O & G production (true). It then goes on to state that light alkanes are often co-emitted in such regions and since propane is a well known tracer for O & G operations, a strong correlation with propane indicates a common source. I suggest reformulating the discussion in this section a little bit. A problem with the reasoning put forward in the first part of the paragraph is that there are other sources that contain all the light alkanes such as auto exhaust and they will correlate with propane and each other because they are co-emitted from this source also. So I would discuss Figures 3a and 3b more holistically  - in other words taken by itself Figure 3a doesn't really have a great deal of meaning – however combined with 3b it does. So I would just rewrite this section a bit to more clearly make the point. I would also encourage the authors to look at expected relative and absolute abundance of NMHCs in the North Sea vs an urban area in the UK – this could be a good metric for establishing the influence of O &G operations on the region. The authors have already done this analysis (Fig 7) for each field type – just need to compare to an urban area. The relative abundance of the light alkanes compared to other NMHC species should really show up in such an analysis. Depending on the outcome of this analysis, some of the other sections that further explore the evidence that the observed emissions are from O & G operations could be shortened

Statements such as: "Aromatic compounds benzene and toluene were also well correlated with propane in West Sheltand, possibly representative of the fact that this region is dominated by oil production and hence emissions of higher carbon number species are expected (Warneke et al., 2014)" are what I referred to earlier as speculative. Not sure this reasoning is solid either. Another explanation is that the source is combustion and not O & G. This is where I think you can make some good points by looking at the absolute concentrations of propane to these compounds and compare to an urban area.

The strongest argument in this section is found in this sentence:  "In general, much weaker correlations of all species with acetylene (compared to propane) were observed in all regions (Fig. 3b and Table 1). Particularly weak correlations were seen in West Shetland (-0.07 < r < 0.53), supporting the conclusion

that O&G activities were the dominant source of VOC emissions in this region with little influence from other sources"

**The discussion of the benzene to toluene ratio**: be consistent with discussing the B/T ratio and don't switch back and forth between B/T and T/B as it adds unnecessary confusion. I found the significant positive correlations interesting – especially when combined with the high B/T ratios. I am not aware of previous studies showing the relatively tight correlations of these species attributable to O & G operations – perhaps point this out or if there are other studies that have shown this then reference them. The tight correlations are interesting in general because of the disparate lifetimes of the two species. I suggest adding more discussion on this. I would discuss this in the context of expected transport times. In the discussion of Figure 4b the bit of highly correlated data with B/T > 4 is unusual and the sentence describing it is in need of some rework. It states:

"… implying evidence of either an aged emission source due to the high proportion of benzene relative to toluene or an additional source of benzene that is not co-emitted with toluene"

-if it was an aged emission then it would not correlate strongly and if the benzene is not co-emitted with toluene it also would not be correlated strongly. So needs more – and better – explanation. To get ratios of B/T that high starting from a traffic emission source would take 6 days ish of aging which would lead to essentially no correlation.

**OH reactivity section**: …Fig. 7b indicates that in the more remote offshore environment, where there are significantly less emission sources and VOC concentrations are generally lower, OH reactivity is dominated by fast reacting species with OH. This is an interesting conclusion. Perhaps it shows that a) the emissions from the North Sea O&G operations are relatively well controlled and b) urban areas in the region are still responsible for a large the VOC reactivity in the region. If this is a conclusion, I think it should be stated.

**Comparison with inventories**
The comparisons are interesting but they don't ultimately inform on the accuracy of the NAEI source profiles – as stated.

**Other:**

Page 3 line 25: Provide urls for the different projects

Page 4 line 10: Nowegian misspelled

Page 5: **Laboratory analysis of VOCs**
Provide more information on analytical specifications – particularly uncertainty in analyses. This doesn't have to be long but to give a sense of whether we are talking +/- 5% or +/-50% and if there is a drop off in precision for the higher MW species on the PLOT column. Source of the calibrations standard, etc.

Page 6 lines 7 and line 9 – units should be consistent and SI

Page 6 line 11: WAS was defined earlier as Whole Air Sampling which doesn't make total sense here.

Page 6 line 25: instead of referencing Hong et al., 2019, it would be better to say: (at $[OH]_{avg}$ = 1.3E06 and T = 260K).

Page 8 line 4: Table 1 – what are the two Norwegian sectors in the table?

Page 8 line 15: I suggest being consistent with r and $r^2$

Page 12 line 25: ..simple metric to identify key species that most readily from peroxy radicals (Gilman et al., 2013) – this has been known since the 1970s so you should either just cite the earlier source or Gilman and the earlier source.

---

## Author Comment (AC1) · 29 Jan 2021

**Author responses to referee comments of acp-2020-1099 – *Speciation of VOC emissions related to offshore North Sea oil and gas production**

**Shona E. Wilde**[*], Pamela A. Dominutti , Grant Allen , Stephen J. Andrews , Prudence Bateson , Stephane. J-B. Bauguitte , Ralph R. Burton , Ioana Colfescu , James France , James R. Hopkins , Langwen Huang , Anna E. Jones , Tom Lachlan-Cope , James D. Lee , Alastair C. Lewis , Stephen D. Mobbs , Alexandra Weiss , Stuart Young , and Ruth M. Purvis

[*]shona.wilde@york.ac.uk

January 29, 2021

**Editorial changes**

1. Three additional authors added: Grant Allen, Prudence Bateson and Langwen Huang

**Response to reviewers**

The authors thank the reviewers for their positive and constructive comments relating to this paper. We feel the paper is much improved after taking these comments into consideration. We address the specific points of each reviewer below. Reviewer comments are shown in **black**, author response in **blue** and text added or amended in **pink**.

**Reviewer #1**

1. The measurements are of high quality, the analysis is thorough, and the paper is very well written. I suggest publication with only a few minor technical corrections.

   Thank you.

2. Page 15, Line 24. Change "representative" to "represent"

   The manuscript has been updated with this change.

3. Figure 8b: It would be best to change these to colored lines rather than the "fill to zero" as you can't see what is beneath the orange-colored oil curve.

   The figure has been updated with this change (Now Figure 7).

[Figure]

**Figure 7.** (a) Mean excess mole fraction for each offshore field type. Error bars represent one standard error. (b) Smoothed density distribution of water depth obtained for each offshore field in the North Sea, coloured by field type. A depth of 305 m (1000 ft) defines deepwater.

4. It would be helpful if Figures 1 and 5 were combined.

The manuscript has been updated with this change.

[Figure]

**Figure 1.** (a) Regions of the North Sea defined for analysis. The black lines represent the flight tracks of the research aircraft. (b) Location of all offshore fields in the North Sea. Each polygon is coloured by the extraction product from each field. (c) Percentage contribution of each offshore field type to the total area of all fields in each region. Country polygons were obtained using the **rnaturalearth** R package (**?**).

5. It would be helpful if the colors on Figures 1b and 5 were consistent. Additionally, the orange and yellow colors used throughout were often hard to differentiate. Perhaps a darker saturation on the orange color or changing one to a different color would help the reader tell the colors apart more easily.

   As above, Figures 1 and 5 have now been combined and the colour schemes are now consistent. The colour scheme throughout the paper (also demonstrated above) has been enhanced to allow clear differentiation.

6. Did the authors measure cycloalkanes? These are often prevalent in crude oil and raw natural gas.

   Unfortunately cycloalkanes were not quantified as part of this study.

7. Figure 4b and Line 13. I agree that either case could be true, but the slope is more suggestive of an additional benzene source as the mixing ratios for benzene are the largest for nearly all regions studied. Chemical aging would also occur along with dilution but these samples look fairly concentrated by comparison. Were any other VOCs enhanced in the samples with enhanced benzene? It could help point to a more specific source. Benzene is used in glycol dehydrators, a common piece of equipment in US gas fields. I wonder if similar equipment is used in these fields and could be the source of benzene-only emissions other than solvent usage.

   This point is address alongside comments relating to this paragraph from Reviewer #2 (see below).

8. Are the authors planning to calculate emission fluxes of methane and VOCs using these measurements in a future analysis? It would be great if so.

   Yes, analysis is currently ongoing to quantify emission fluxes of $CH_4$ and individual VOCs at a facility-level through application of the mass balance technique.

**Reviewer #2**

1. The paper is well written and a large amount of high quality work went into conducting the flights, analytical work, data analysis, and synthesis and I applaud the author's efforts.

   Thank you.

2. I like Figure 2 as it nicely shows that the four different sectors have i-pentane to pentane ratios that are indicative of oil and gas operations influence. Somewhere I would state that emissions from sources other urban and O&G are not expected in this region (e.g., fires).

   The following sentence has been added to the text:

   Additional sources of emissions such as biomass burning are not expected to influence VOC concentrations in this region, therefore the $iC_5/nC_5$ ratio is a suitable tool for the differentiation of urban and O&G emissions.

3. In section 3.1.2 there is a discussion about correlations with tracer compounds. Here in a lead up to discussing Figure 3 it is stated that mixing ratios of propane are often elevated in regions of O&G production (true). It then goes on to state that light alkanes are often co-emitted in such regions and since propane is a well known tracer for O&G operations, a strong correlation with propane indicates a common source. I suggest reformulating the discussion in this section a little bit. A problem with the reasoning put forward in the first part of the paragraph is that there are other sources that contain all the light alkanes such as auto exhaust and they will correlate with propane and each other because they are co-emitted from this source also. So I would discuss Figures 3a and 3b more holistically - in other words taken by itself Figure 3a doesn't really have a great deal of meaning — however combined with 3b it does. So I would just rewrite this section a bit to more clearly make the point. I would also encourage the authors to look at expected relative and absolute abundance of NMHCs in the North Sea vs an urban area in the UK — this could be a good metric for establishing the influence of O&G operations on the region. The authors have already done this analysis (Fig 7) for each field type — just need to compare to an urban area. The relative abundance of the light alkanes compared to other NMHC species should really show up in such an analysis. Depending on the outcome of this analysis, some of the other sections that further explore the evidence that the observed emissions are from O&G operations could be shortened

   This section has now been refined and re-drafted as suggested, including analysis regarding the relative abundance of NMHCs. The section now reads:

[revised manuscript text omitted]

4. The discussion of the benzene to toluene ratio: be consistent with discussing the B/T ratio and don't switch back and forth between B/T and T/B as it adds unnecessary confusion.

The sentence has been amended and now reads:

These values are consistent with findings from a study at BAO by ? who found the highest benzene–toluene ratios in the north-east (B/T = 1.32 ± 0.25) sector were attributable to O&G emissions.

5. I am not aware of previous studies showing the relatively tight correlations of these species attributable to O&G operations – perhaps point this out or if there are other studies that have shown this then reference them.

The tight correlations are only indicative of a common emission source. The nature of the source is determined from the slope of the linear fit between the two species as stated already in the text.

6. The tight correlations are interesting in general because of the disparate lifetimes of the two species. I suggest adding more discussion on this. I would discuss this in the context of expected transport times.

More discussion has been added in the following paragraph:

Figure 4b shows the regression plot of benzene versus toluene for the South UK coloured by wind direction sector. Data with a B/T ratio between 0.41–0.83 (traffic emissions) is plotted with a diamond, accounting for 3.5 % of the observations in the Southern region. A strong positive correlation was found to exist (r(14) = 0.92, p<.001) for the traffic source and the slope obtained from the linear fit was 0.60, in the centre of the range expected for vehicle emissions (?). The traffic source was primarily observed when the wind direction was from the south or south-west, suggestive of air transported from the UK or from continental Europe polluted by urban vehicular emissions. A similar traffic source is also visible in the North UK data, similarly exclusively observed under southerly wind conditions. Air transported from the UK mainland or Europe is expected to reach the location of the aircraft flight tracks in less than a day. The estimated lifetimes for benzene and toluene are 2 weeks and 2 days respectively (?), meaning emissions of benzene and toluene from urban areas are expected to remain well correlated on this relatively short spatial scale since they are also similarly affected by dilution and mixing.

7. In the discussion of Figure 4b the bit of highly correlated data with B/T > 4 is unusual and the sentence describing it is in need of some rework. It states: … implying evidence of either an aged emission source due to the high proportion of benzene relative to toluene or an additional source of benzene that is not co-emitted with toluene" –if it was an aged emission then it would not correlate strongly and if the benzene is not co-emitted with toluene it also would not be correlated strongly. So needs more – and better – explanation. To get ratios of B/T that high starting from a traffic emission source would take 6 days ish of aging which would lead to essentially no correlation.

This has been addressed alongside comments made by Reviewer #1. This paragraph has now been re-drafted in the manuscript following additional analysis.

There is a section of highly correlated data (r(214) = 0.81, p<.001) characterised by B/T ratios > 4 (squares, Fig. 4b). High B/T ratios can be indicative of aged emissions due to the high proportion of benzene relative to toluene. However, given that the benzene mixing ratios observed were among the largest for all the regions studied, this is more suggestive of an additional benzene source from the offshore platforms. Other compounds related to fossil fuel combustion and evaporation such as ethene and pent-1-ene were also enhanced in these samples, providing evidence of an additional combustion source enriched in benzene. There are a host of potential sources on drilling rigs including gas turbines, which are widely used for power generation and compressors, both of which have previously been linked to elevated benzene concentrations (??).

8. OH reactivity section: …Fig. 7b indicates that in the more remote offshore environment, where there are significantly less emission sources and VOC concentrations are generally lower, OH reactivity is dominated by fast reacting species with OH. This is an interesting conclusion. Perhaps it shows that a) the emissions from the North Sea O&G operations are relatively well controlled and b) urban areas in the region are still responsible for a large the VOC reactivity in the region. If this is a conclusion, I think it should be stated.

In regard to (a), we cannot speculate on how well emissions from O&G operations are controlled based on this figure. The fact that OH reactivity is not controlled by emissions of alkanes relates to the relatively low concentrations measured in the North Sea. Generally in onshore US O&G fields, concentrations of VOCs are much due to the high density of wells and equipment on the well pads. Therefore, the overwhelmingly high emissions of light alkanes is enough to offset the difference in reactivity compared to unsaturated species, meaning OH reactivity is dominated by alkanes. As stated here, we measured generally much lower concentrations of all species and therefore unsaturated compounds dominate the reactivity.

In regard to (b), the atmospheric lifetime of species such as ethene and 1,3-butadiene are short, meaning there is generally limited potential for long-range transport of these compounds. Urban concentrations of 1,3-butadiene are typically less than 0.5 ppb, therefore it is unlikely that fast reacting species such as these would be present in similar concentrations over the North Sea if the source was from urban centres on the UK mainland. More likely is that these species are emitted from the more general combustion sources that exist on oil and gas platforms.

The text has been updated to read:

However, Fig. 6b indicates that in the more remote offshore environment, where there are significantly less emission sources and VOC concentrations are generally lower, OH reactivity is dominated by fast reacting species with OH. These are likely emitted as a result of the more general combustion sources that exist on O&G platforms.

9. Page 3 line 25: Provide urls for the different projects

   The manuscript has been updated with this change.

10. Comparison with inventories – The comparisons are interesting but they don't ultimately inform on the accuracy of the NAEI source profiles – as stated.

    Whilst we recognise that this data set is not ideal for comparison to the inventory speciation profiles, we feel it is important to show how these emissions are currently represented. Our comparison exposed the simplicity of some of the profiles, suggesting that emissions of some species may not be fully accounted for.

11. Page 4 line 10: Nowegian misspelled

    The manuscript has been updated with this change.

12. Page 5: Laboratory analysis of VOCs – Provide more information on analytical specifications – particularly uncertainty in analyses. This doesn't have to be long but to give a sense of whether we are talking +/- 5% or +/-50% and if there is a drop off in precision for the higher MW species on the PLOT column. Source of the calibrations standard, etc.

    This section has been amended and the text now reads:

    Peak identification was made by reference to a National Physical Laboratory (NPL) calibration gas standard containing a known amount of 30 non-methane hydrocarbons (NMHCs) ranging from $C_2$–$C_9$ (NPL30, D600145– 2018, UK). Peak integration, blank correction and the application of calibration data to calculate mixing ratios was conducted using GC Soft, Inc software. VOC extended uncertainties during our analyses ranged from 0.72% to 8.37%, with higher values obtained for octane (5.80%), acetylene (6.61%) and toluene (8.37%). Uncertainties were calculated based on the accuracy and precision resulting from multiple injections of the standard into the GC-FID and were multiplied by a coverage factor of 2 to give 95% confidence intervals.

13. Page 6 lines 7 and line 9 – units should be consistent and SI

    Units are now consistently in m. The sentence has been amended and now reads:

    PBL profiles were generally conducted upwind of the area of interest and typically spanned an altitude range of 15 m–1500 m.

14. Page 6 line 11: WAS was defined earlier as Whole Air Sampling which doesn't make total sense here.

    The sentence has been amended and now reads:

    Canister samples were captured solely within the PBL on each flight.

15. Page 6 line 25: instead of referencing Hong et al., 2019, it would be better to say: (at [OH]avg = 1.3E06 and T = 260K).

    The manuscript has been updated with this change.

16. Page 8 line 4: Table 1 – what are the two Norwegian sectors in the table?

    This was a formatting error and has now been corrected.

17. Page 8 line 15: I suggest being consistent with r and r2

    Pearson correlation coefficients (r) have been added to the table to make it consistent with the discussion in the text.

18. Page 12 line 25: ..simple metric to identify key species that most readily from peroxy radicals (Gilman et al., 2013) – this has been known since the 1970s so you should either just cite the earlier source or Gilman and the earlier source.

    The citation has now been updated.